# Uncertainty Quantification for Deep Regression using Contextualised Normalizing Flows

**Adriel Sosa Marco**[‡], **John Daniel Kirwan**[‡], **Alexia Toumpa**[§], **Simos Gerasimou**[§*]

[‡]Arquimea Research Center, Spain
[§]Department of Computer Science, University of York, York, UK
[*]Department of Elect. Eng., and Computer Science and Eng., Cyprus University of Technology, Cyprus
{asosa, jkirwan}@arquimea.com
{alexia.toumpa,simos.gerasimou}@york.ac.uk

## Abstract

Quantifying uncertainty in deep regression models is important both for understanding the confidence of the model and for safe decision-making in high-risk domains. Existing approaches that yield prediction intervals overlook distributional information, neglecting the effect of multimodal or asymmetric distributions on decision-making. Similarly, full or approximated Bayesian methods, while yielding the predictive posterior density, demand major modifications to the model architecture and retraining. We introduce MCNF, a novel post hoc uncertainty quantification method that produces both prediction intervals and the full conditioned predictive distribution. MCNF operates on top of the underlying trained predictive model; thus, no predictive model retraining is needed. We provide experimental evidence that the MCNF-based uncertainty estimate is well calibrated, is competitive with state-of-the-art uncertainty quantification methods, and provides richer information for downstream decision-making tasks.

## 1 Introduction

Deep regression models have been widely used in applications involving the prediction of continuous variables [1], including drug discovery [2], credit scoring [3] and energy forecasting [4]. Despite their broad adoption, safety-critical applications, like medical diagnostics [5], entail high-stake decisions, mandating the development of robust deep regression techniques that equip decision-makers with complementary knowledge about the predictive uncertainty of a regression model.

Uncertainty quantification methods (UQ) are fundamental in establishing the predictive uncertainty of regression models [6]. Recent advances target primarily the investigation of *epistemic* uncertainty (caused by the lack of evidence or knowledge during training) and *aleatoric* uncertainty (the irreducible uncertainty due to data stochasticity). Building on the foundational work in quantile regression [7], Monte Carlo Dropout (MCD) [8] and deep ensembles [9] leverage dropout layers and multiple functionally-equivalent models, respectively, to approximate at inference time a deep Gaussian process and estimate the predictive distribution. Both methods, however, incur extra computational overheads and suffer from inefficient sampling, particularly at the predictive distribution tails [10]. Bayesian-based approaches that perform full Bayesian estimation [11] or its variational counterpart [12], albeit rigorous in uncertainty incorporation in the posterior distribution, incur prohibitive computational costs for any modern deep learning model. Conformal prediction (CP) [13] yields statistically rigorous uncertainty intervals that contain the ground truth based on a user-defined error rate [14], but its reliance on a calibration dataset restricts its applicability.

Motivated by the need for rigorous UQ in safety-critical applications, we introduce MCNF, a post hoc distribution-free UQ method for deep regression models underpinned by contextualized normalizing

flows (NF). MCNF uses a trained deep regression model equipped with dropout layers and yields statistically rigorous uncertainty intervals arising from the full predictive density function. Under the hood, MCNF exploits MCD sampling more efficiently to produce a prediction set per input leveraging key statistical information (e.g., mean, variance) to condition (contextualize) the normalizing flow [15]. Our experimental evaluation using a diverse set of datasets and state-of-the-art UQ methods [14, 16] demonstrates that MCNF achieves competitive results in terms of marginal coverage while also having lower error values and narrower intervals. Its applicability to deep learning architectures other than feed-forward regression networks is also showcased. Similarly to CQR [14] and MCCP [16], MCNF operates at inference time while also being capable of representing arbitrarily complex uncertainty distributions, which neither CQR nor MCCP support. Our concrete contributions are:

• The MCNF method for the uncertainty quantification whose estimates are in the form of a distribution-agnostic predictive distribution.

• A comprehensive MCNF evaluation against state-of-the-art UQ methods (MCD, CQR, MCCP) on various standard benchmarks and a physicochemical dataset.

• A prototype open-source MCNF tool and case study repository, available at `https://github.com/alexiatoumpa/MCNF`.

## 2  Related Work

Uncertainty Quantification (UQ) in deep regression models remains an open-ended question [10]. Quantifying uncertainty in deep learning models enables reasoning about the model's confidence in its predictions. Quantile Regression (QR) [7] constructs prediction intervals by modeling the relationship between a set of independent variables and quantiles of target variables. A typical approach for UQ is training Bayesian Neural Networks (BNNs) [17], comprising neural networks with a probability distribution for the model parameters that learn the predictive posterior of the target variable. Although the output probability distribution of a BNN captures the model uncertainty, BNNs are computationally-intensive, demanding significantly more training time than other methods [11].

Monte Carlo Dropout (MCD) [8] samples weights in each layer using a binomial distribution at the selected dropout rate. Each forward pass produces a new estimate coming from the predictive posterior, which approximates Bayesian inference of a Gaussian process [18]. This technique draws parallels to deep ensembles [9], but shares weights across model realizations.

Variational inference approximates full Bayesian approaches by introducing a family of distributions that make the modeling problem tractable, commonly amortizing the parameters of the posterior [19] with an auxiliary function trained on the data.

Conformal Prediction (CP) [13, 20] is a distribution-free, non-parametric forecasting method which exploits past experience to determine the level of confidence for new predictions. CP produces prediction intervals indicating the confidence level of the model, which is inversely-related to the interval size [21]. Conformal Quantile Regression (CQR) [14] combines quantile regression with conformal prediction techniques, aiming to construct prediction intervals without distributional assumptions. MCCP [16] combines Monte Carlo dropout and conformal prediction techniques by dynamically adapting the conventional MCD with a convergence condition and employing advanced conformal prediction techniques for the synthesis of robust prediction intervals.

## 3  Preliminaries

**Normalizing Flows** (NF) [22, 23] is a modeling framework for the characterization of arbitrarily complex probability distributions, often referred to as target distribution $p_{\mathbf{X}}(\mathbf{x})$, where a set of invertible and differentiable transformations is applied over simple probability density functions (e.g., uniform, standard normal) or a base distribution $p_{\mathbf{Z}}(\mathbf{z})$. NF samples can be drawn from the target distribution by sampling from the base distribution and applying the set of transformations that convert the latent variable $\mathbf{Z}$ into the original random variable $\mathbf{X}$, and estimating the density for a given value of the random variable $\mathbf{X}$.

Let $\mathbf{X} \in \mathbb{R}^d$ be a random variable with an intractable probability density function and $Z \in \mathbb{R}^d$ another random variable with a known and tractable probability density function. Let also $\mathbf{g} = g_1 \circ \cdots \circ g_L$ be a set of $L$ differentiable and invertible functions composition (i.e., a flow) such that $\mathbf{X} = \mathbf{g}(\mathbf{Z})$.

The density function of the target random variable can be expressed in terms of the base density by:

$$p_{\mathbf{X}}(\mathbf{x}) = p_{\mathbf{Z}}(\mathbf{g}^{-1}(\mathbf{x})) \prod_{l=1}^{L} \left| \det \left( \mathrm{J} g_l(g_l^{-1}(\mathbf{x})) \right) \right|^{-1} \tag{1}$$

Since NFs can estimate the density function, they can be trained by maximizing the likelihood with respect to the parameters of the transformation functions of the given data [15].

**Neural Spline Flows** (NSF) [24] are a type of transformation flow that fulfills the NF requirements of invertibility and differentiability [15]. Monotonic rational-quadratic splines endow the transformations with high non-linearity and flexibility, where the support vector (or knot) widths and heights and the derivatives of the polynomial on the internal knots are estimated using a neural network with learnable parameters. These transformations can easily integrate conditions to model conditioned density functions $p_{\mathbf{X}}(\mathbf{x}|\mathbf{c})$, where $\mathbf{c}$ is the condition (or context) vector.

## 4   MCNF

**M**onte **C**arlo **N**ormalizing **F**low (MCNF), whose high-level workflow is shown in Fig. 1, enables quantifying the predictive uncertainty for regression tasks. Let $\mathcal{D} = \{\mathbf{x}, y\}^N$ denote a dataset of size $N$ for a regression task, where $\mathbf{x}$ and $y$ are the predictor vector and the predicted variable, respectively. Then, in MCNF, the predictive posterior distribution $p(y|\mathbf{x}, \mathcal{D})$ resulting from the Monte Carlo Dropout (MCD) [8] is calibrated post hoc and is fully decoupled from the training of the predictive (base) model that describes the data. Thus, MCNF separates the modeling task from UQ, and employs MCD to derive prior estimations, leveraging the commonly used *dropout* as a regularization technique in deep learning architectures.

MCNF, like CP [14], estimates $p(y|\mathbf{x}, \mathcal{D})$ without making distributional assumptions about the uncertainty. Specifically, we achieve this distribution-free concept by applying Normalizing Flow (NF) [15] as a downstream post-processing of the MCD-generated samples. Rather than modeling the predictive variable directly, MCNF uses the NF to describe the distribution of the prediction errors conditioned on the prior prediction, propagating the epistemic uncertainty encoded by the MCD prior estimate and combining it with the aleatoric uncertainty of the underlying process.

### 4.1   Formal description

MCNF uses estimates from MCD ($y_{\mathrm{MCD}}$) as a latent variable acting as a prior to the actual predictive distribution. Thus, we express the predictive probability distribution $p(y|\mathbf{x}, \mathcal{D})$ by marginalizing the joint probability distribution $p(y, y_{\mathrm{MCD}} | \mathbf{x}, \mathcal{D})$ over the prior.

$$p(y|\mathbf{x}, \mathcal{D}) = \int p(y, y_{\mathrm{MCD}} | \mathbf{x}, \mathcal{D}) \, dy_{\mathrm{MCD}} = \int p(y | y_{\mathrm{MCD}}, \mathbf{x}, \mathcal{D}) p(y_{\mathrm{MCD}} | \mathbf{x}, \mathcal{D}) \, dy_{\mathrm{MCD}}$$
$$= \mathbb{E}_{p(y_{\mathrm{MCD}} | \mathbf{x}, \mathcal{D})} \left[ p(y | y_{\mathrm{MCD}}, \mathbf{x}, \mathcal{D}) \right] \tag{2}$$

Therefore, the predictive distribution is expressed in the form of a mathematical expectation of the conditional distribution $p(y|y_{\mathrm{MCD}}, \mathbf{x}, \mathcal{D})$. Since Equation (2) is intractable, we resort to Monte Carlo approximation to estimate the predictive distribution of $y$, resulting in:

$$p(y|\mathbf{x}, \mathcal{D}) = \mathbb{E}_{p(y | y_{\mathrm{MCD}}, \mathbf{x}, \mathcal{D})} \left[ p(y | y_{\mathrm{MCD}}, \mathbf{x}, \mathcal{D}) \right] \approx \frac{1}{n_{\mathrm{MCD}}} \sum_{i=1}^{n_{\mathrm{MCD}}} p(y | y_{\mathrm{MCD}}^{(i)}, \mathbf{x}, \mathcal{D}) \tag{3}$$

where $n_{\mathrm{MCD}}$ is the number of prior samples to approximate the predictive distribution.

To account for the epistemic uncertainty propagation from the prior to the predictive distribution, we introduce the change of variable, $\delta = y - y_{\mathrm{MCD}}$, which reflects the prediction error. Hence, the conditioned probability distribution is now expressed in terms of $\delta$ instead of $y$, given by $p(\delta|y_{\mathrm{MCD}}, \mathbf{x}, \mathcal{D})$. This change does not affect the calculation of Equation (2) nor Equation (3), due to the linearity of the transformation, so $p(y | y_{\mathrm{MCD}}, \mathbf{x}, \mathcal{D}) = p(\delta | y_{\mathrm{MCD}}, \mathbf{x}, \mathcal{D})$. Therefore, Equation (3) can be expressed in terms of $\delta$ as:

$$p(y|\mathbf{x}, \mathcal{D}) \approx \frac{1}{n_{\mathrm{MCD}}} \sum_{i=1}^{n_{\mathrm{MCD}}} p(\delta_i | y_{\mathrm{MCD}}^{(i)}, \mathbf{x}, \mathcal{D}) \tag{4}$$

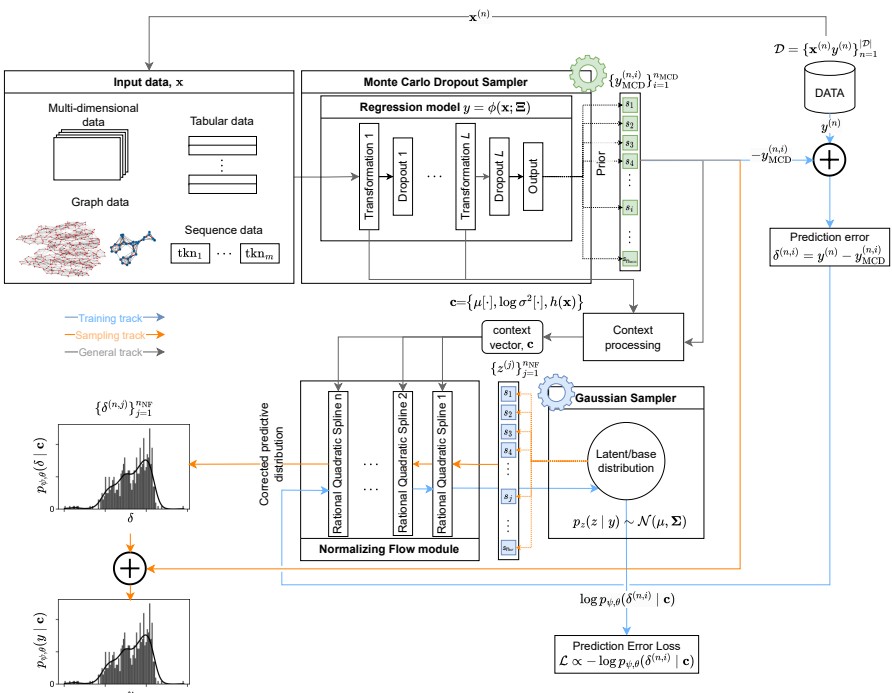

Figure 1: Overview of the proposed MCNF UQ method for regression tasks. First, a set of samples is drawn using Monte Carlo Dropout (MCD). These samples are used to build a context vector that encodes the MCD predictive posterior and, ultimately, the input observation using a convenient set of summary statistics. These summary statistics are provided to the Normalizing Flow-based model as a context. Depending on the task, either a requested number of samples can be drawn from the Normalizing Flow by sampling the base distribution and submitting those samples through the forward pass, or the likelihood of the input observations can be assessed.

Accordingly, the prior distribution can be approximated using the Monte Carlo Dropout sampling process, while the conditioned distribution of the prediction errors, $\delta$, is modeled through the normalizing flow, $\mathcal{F}^{-1}\left(\delta, y_{\mathrm{MCD}}, \mathbf{x}\right) \sim p_{\boldsymbol{\theta}, \boldsymbol{\psi}}\left(\delta \mid y_{\mathrm{MCD}}, \mathbf{x}, \mathcal{D}\right)$ providing an efficient correction of the prior. In the latter expression, $\boldsymbol{\theta}$ and $\boldsymbol{\psi}$ are, respectively, the flow transformation parameters and the base distribution parameters.

## 4.2 Building the MCNF Context

In MCNF, the prediction errors are conditioned on the prior estimate $y_{\mathrm{MCD}}$ and the observed input features (or descriptors) $\mathbf{x}$. These two variables configure a context vector $\mathbf{c}$ that should be fed to the normalizing flow head of MCNF as a context to locate and scale the predictive distribution. Through this contextualization, MCNF defines a convenient procedure so that the context definition is agnostic to the input data structure, and in such a way that MCNF can still be applied post hoc. To accomplish this, we define a proxy, $h(\mathbf{x})$, over the input descriptors based on an internal representation of the regression model, $\phi$, with its learnt parameters, $\Xi$, $\phi(\mathbf{x}; \Xi)$. This approach leverages the feature extraction made by the regression model and uses it to define a proper context for the normalizing flow. Since one or more layers may be selected at different depths, this becomes a tunable hyperparameter of MCNF that directly impacts the dimensionality of the context vector. This procedure normalizes the shape of the proxy, which helps to generalize the method.

The context vector should be completed by virtue of the latent variable: the prior estimate $y_{\mathrm{MCD}}$. In practice, we observed that achieving numerical stability and computational efficiency entailed replacing each sample generated using MCD by the estimates of the two first moments of the prior

distribution. Thus, the context vector is expressed as follows:

$$\mathbf{c} = \left\{ \bar{y}_{\text{MCD}}, \log s^2 \left( \bar{y}_{\text{MCD}} \right), h(\mathbf{x}) \right\} \tag{5}$$

where $\bar{y}_{\text{MCD}}$ and $\log s^2 \left( \bar{y}_{\text{MCD}} \right)$ are the sample estimates of the expectation and variance of the prior distribution.

Finally, we adapt the definition of the density function of the predictive distribution described in Equation (4) according to the proposed context vector (Equation (5)) to obtain our working equation.

$$p_{\boldsymbol{\theta}, \boldsymbol{\psi}}(y|\mathbf{x}, \mathcal{D}) \approx \frac{1}{n_{\text{MCD}}} \sum_{i=1}^{n_{\text{MCD}}} p_{\boldsymbol{\theta}, \boldsymbol{\psi}}(\delta_i \,|\bar{y}_{\text{MCD}}, \log s^2 \left( \bar{y}_{\text{MCD}} \right), h(\mathbf{x}), \mathcal{D}) \tag{6}$$

Note that Equation (6) is not equivalent to Equation (4). However, considering the hypothesis of normality of the distribution estimated with MCD, we assume that using the first two moments of the distribution of the latent random variable allows us to obtain a reasonable approximation of the marginalized distribution.

## 4.3  MCNF Training

MCNF training is based on the forward Kullback-Leibler divergence $D_{\text{KL}}$ (Equation (7)) [15]. This is equivalent to minimizing the negative log-likelihood of the training data with respect to the model parameters $\boldsymbol{\theta}$ (for the transformation flows) and $\boldsymbol{\psi}$ (for the base distribution).

$$\begin{aligned} \mathcal{L}_{\text{NL}}(\boldsymbol{\theta}, \boldsymbol{\psi}) &= D_{\text{KL}} \left[ p_{y|\mathbf{x}}(y|\mathbf{x}) \| p_{\boldsymbol{\theta}, \boldsymbol{\psi}}(y|\mathbf{x}, \mathcal{D}) \right] \\ &= -\mathbb{E}_{p_{y|\mathbf{x}}(y|\mathbf{x})} \left[ \log p_{\boldsymbol{\theta}, \boldsymbol{\psi}}(y|\mathbf{x}, \mathcal{D}) \right] + \text{const.} \\ &\approx -\frac{1}{N} \sum_{n=1}^{N} \log p_{\boldsymbol{\theta}, \boldsymbol{\psi}} \left( g^{-1}(y_n; \mathbf{c}, \boldsymbol{\theta}) \right) + \log \left| \det J_{g^{-1}}(y_n; \mathbf{c}, \boldsymbol{\theta}) \right| + \text{const.} \end{aligned} \tag{7}$$

It is well established that minimizing the log-likelihood may lead to model overfitting and deformed distributions. This leads to uncalibrated uncertainty estimates, which become more apparent for low-uncertainty effects corrupted by large outliers. To rule this out, we regularize Equation (7) by weighing observations proportionally to the prior density. Since MCNF provides post hoc corrections of the MCD predictive posterior, we use this as the prior and reformulate $\mathcal{L}_{\text{NL}}(\boldsymbol{\theta}, \boldsymbol{\psi})$ as:

$$\mathcal{L}_{NL}(\boldsymbol{\theta}, \boldsymbol{\psi}, \tau) = -\sum_{n=1}^{N} \mathbf{w}_n \left( \log p_{\theta, \psi} \left( g^{-1}(y_n; \mathbf{c}_n, \boldsymbol{\theta}) \right) + \log \left| \det J_{g^{-1}}(y_n; \mathbf{c}_n, \boldsymbol{\theta}) \right| \right) \tag{8}$$

$$\text{where } \mathbf{w}_n = \sigma \left( -\frac{\log p_{\text{MCD}}(y_n|\mathbf{x}_n)}{\tau}; \tau \right) \in (0, 1)$$

By introducing $\mathbf{w_n}$ in Equation (8), we scale the importance of each observation in a mini-batch according to its departure from the prior. We use a softmax function $\sigma(\cdot)$ such that weights fall within the (0,1) interval. To mitigate the effect of a highly informative prior, as for MCD, we temperature scale the softmax function using a hyperparameter $\tau$. Accordingly, the effect of outliers can be mitigated without affecting the fitting of the regular data. We note that Equation (8) reduces to Equation (7) when $\tau \to \infty$ (reflecting a design decision when there is knowledge that outliers are absent from the data).

Mini-batches are constructed by resampling the MCD samples to account for the variability encoded by the prior. Further details are provided in Algorithm 2 in the Appendix.

## 4.4  MCNF Inference

MCNF operates hierarchically, i.e., samples need to be drawn from the predictive model using MCD to generate the context prior to assessing the predictive distribution of the uncertainty using the Normalizing Flow head. This means that to generate $n^{\text{NF}}$ samples from the approximated predictive distribution $p_{\boldsymbol{\theta}, \boldsymbol{\psi}}(y|\mathbf{x}, \mathcal{D})$, we first need to generate $n_{\text{MCD}}$ samples from the base model $\phi(\mathbf{x}, \Xi)$. We will refer to this set of samples as the *prior* or *warm-up* samples.

The required input data to make estimates according to Algorithm 1 consist of the number of prior samples $n_{\text{MCD}}$, as well as the architecture of the Normalizing Flow $\mathcal{F}^{-1}(\delta, \mathbf{c})$ and the number of

---

**Algorithm 1** Steps involved in a full forward pass of the proposed MCNF method for inference

---

**Input**: $\mathcal{D} = \{\mathbf{x}_n, y_n\}_{n=1}^{N_{\text{Test}}}$
**Parameters**: $n_{\text{MCD}}$, $\phi(\mathbf{x}, \Xi)$, $\mathcal{F}(\delta, \mathbf{c})$, $n_{\text{NF}}$
**Output**: $y_{\text{pred}}$ and/or $p_{\boldsymbol{\theta}, \boldsymbol{\psi}}(y|\mathbf{x}, \mathcal{D})$

1: Run $n_{\text{MCD}}$ forward passes of $\phi(\mathbf{x}_n, \Xi)$ times using MCD for every every observation, $\mathbf{x}_n$.
2: Collect hidden states, $h(\mathbf{x}_n)$, generated in each forward pass.
3: Aggregate hidden states from the forward passes.
4: Generate context vector, $\mathbf{c}_n$ according to (5).
5: Feed $\mathbf{c}_n$ to $\mathcal{F}(\delta, \mathbf{c}_n)$
6: Draw $n_{\text{NF}}$ samples, $\delta_n = \{\delta_{k,n}\}_{k=1}^{n_{\text{NF}}}$, from $z_{k,n} \sim p_{0,\boldsymbol{\psi}}(Z) \rightarrow \delta_{k,n} = \mathcal{F}(z_{k,n}, \mathbf{c}_n)$
7: Estimate $y_n$ as $y_n = y_{n,j(k),\text{MCD}} - \delta_{n,k}$, where $j(k) \sim \mathcal{U}(1, n_{\text{MCD}})$
8: Estimate each sample density feeding $\mathcal{F}^{-1}(\delta_{n,k}, \mathbf{c}_n)$ into (6)
9: **return** $\{y_{n,k}\}_{k=1}^{n_{\text{NF}}}$ and $\{p_{\boldsymbol{\theta}, \boldsymbol{\psi}}(y_{k,n}|\mathbf{x}_{k,n}, \mathcal{D})\}_{k=1}^{n_{\text{NF}}}\}_{n=1}^{|\mathcal{D}|} \in \mathbb{R}^{|\mathcal{D}| \times n_{\text{NF}}}$

---

samples requested to characterize the approximate predictive distribution $n_{\text{NF}}$. The inference process starts by building the context $\mathbf{c}$. This entails drawing the requested $n_{\text{MCD}}$ samples from the regression model $\phi(\mathbf{x}, \Xi)$ using MCD, which are then summarized to generate the sample mean and log-var to partially construct the context. Since every forward pass on $\phi(\mathbf{x}, \Xi)$ produces one proxy of the input, we average these estimates to complete the context vector $\mathbf{c}$. Then, the context vector is given as an input to $\mathcal{F}^{-1}(y, \mathbf{c})$ (line 5). For sampling tasks, we first generate samples from the NF base distribution $z_{n,k}$ that are then submitted through the forward pass to obtain $\delta_{n,k} = \mathcal{F}(z_{n,k}, \mathbf{c}_n)$ and then correct the prior $y_{n,j(k),\text{MCD}}$ with the sampled prediction error to finally get $y_{n,k}$ (line 8). During the same forward pass, the Jacobian of the transformation flows is also assessed, enabling likelihood estimation by solving Equation (1) for $p_{\boldsymbol{\theta}, \boldsymbol{\psi}}(y_{k,n}|\mathbf{x}_{k,n}, \mathcal{D})$. For density estimation tasks, an observation $y_n$ is passed along with the context vector $\mathbf{c}_n$. Then, the likelihood is calculated by running the reverse pass of the NF (Equation (1)).

## 5  Evaluation

**Base Predictive Model.** The base predictive model is a Deep Quantile Regressor (DQR) comprising a batch normalization input layer, two fully connected layers (with ReLU nonlinearities and dropout layers with rate 0.1) and an output layer with three linear units for the quantiles $q = \{0.05, 0.5, 0.95\}$. The model is trained for 100 epochs using the Adam optimizer, a custom pinball loss function that aggregates the quantile errors, and a batch size of 32. We fixed the learning rate to 5e-4 and the weight decay regularization factor to 1e-6. For the training and testing sets, we use an 80:20 split.

**Probabilistic Model.** The Normalizing Flow component of MCNF uses a sequence of two Neural Splines flows [24], with a 3-layered multilayer perception comprising 64 hidden units which produces the 16 support vectors of the spline transformation and their inner derivatives. As a base distribution, we use a factorized Gaussian distribution with trainable parameters $\boldsymbol{\psi}$. The context size is determined by the size of the input proxy plus the two statistics (sample mean, and log-var) used to summarize the MCD samples, i.e., $(|h(\mathbf{x})| + 2)$. To keep the computational overhead related to the prior Monte Carlo Dropout sampling low, we set $n_{\text{MCD}} = 50$. The training includes the same partition as the base predictive model, using a batch size of 32 with the Adam optimiser and a 0.001 learning rate. We set $\tau = 1e10$ to instantiate Equation (8), giving the same weight to all observations in a mini-batch.

**Comparative Methods.** We assessed MCNF against five state-of-the-art UQ methods that also use the predictions from the base predictive DQR model for uncertainty estimations. Thus, MCQR involves resampling 1000 times DQR using MCD and averaging the $q = \{0.05, 0.95\}$ quantile outcomes to yield the prediction intervals. MCD derives $p_{\text{MCD}}(y|\mathbf{x})$ by resampling from the median ($q = 0.5$) in DQR 1000 times, resembling the conventional Monte Carlo Dropout uncertainty quantification method. MCD is the only other method that can also produce a predictive distribution. We also used Conformalized Quantile Regression (CQR) [14], which applies a non-conformity score to DQR to conformalize the prediction intervals, and Monte Carlo Conformal Prediction (MCCP) [16], which conformalizes the prediction intervals obtained with MCQR. Both CP-based methods used 20% of the test set for calibration and prediction intervals aimed at achieving a 90% marginal coverage. Note

that for all the considered UQ methods that employ MCD sampling, the aleatoric term of the original formulation [8] is left out as it is used to propagate the epistemic uncertainty only.

**Benchmarks.** Adopting the evaluation procedure from CQR [14] and MCCP [16], we used the following datasets to evaluate MCNF: the Boston Housing dataset [25] (506 observations, 14 attributes); the Concrete dataset [26] (1030 observations, 9 attributes); the Abalone dataset [27] (4177 observations, 11 attributes); the Tertiary Protein Structure dataset [28] (45730 observations, 10 attributes); the wave energy dataset [29] (63600 observations, 149 attributes), and the superconductivity dataset [30] (21263 observations, 81 attributes). These datasets were obtained from [31]. Similar to [14], and to examine the MCNF's ability to capture the uncertainty of complex distributions, we also include two synthetic datasets: the dataset from [14] with univariate predictor samples and few large outliers (termed Romano-Original) and an extension (termed Romano-Mod) with a multimodal distribution.

**Performance Metrics.** To compare the performance of all methods, we used the metrics reported below. We report results across 20 independent runs of the MCNF training procedure described to account for randomness. The training/test partitioning was generated per run and held constant to train both the base predictive model and the comparative methods (MCNF, CQR, MCCP, MCCQR).

• **Marginal coverage** signifies the proportion of the test set over which the predicted quantile intervals include the ground truth, given by $C(X, Y) = \frac{1}{N} \sum_{i=1}^{N} \mathbb{1}\{y_i \in [q_\alpha(x_i), q_{1-\alpha}(x_i)]\}$, where $q_\alpha(x_i)$ and $q_{1-\alpha}(x_i)$ are the predicted quantiles given $x_i$ and $\alpha = 0.05$.

• **Interval size** signifies how well the prediction intervals capture the aleatoric and epistemic uncertainties, with smaller intervals denoting easier inputs and larger intervals harder ones, given by $\tilde{\Delta}(X, Y) = median_{i=1}^{N}(q_{1-\alpha}(x_i) - q_\alpha(x_i))$, where $q_\alpha$, $q_{1-\alpha}$ and $\alpha$ are as above.

• **Accuracy** assesses if the prediction intervals under- or overestimate uncertainty, specified by the mean absolute error (MAE) for $\alpha = 0.5$ and given by $MAE(X, Y) = \frac{1}{N} \sum_{i=1}^{N} |y_i - q_{0.5}(x_i)|$.

## 5.1 Results Summary

**Accuracy and coverage.** Table 1 summarizes the performance results of our evaluation. MCNF overall yielded the smallest MAE, closely followed by MCD, indicating that training the NF MCNF-component helps improve the accuracy of the predicted interval. MAEs for the other methods are larger, usually by one order of magnitude, compared to MCNF and MCD. This is expected since the prediction adjustment carried out in the CP-based methods CQR and MCCP is homogeneous for both upper and lower intervals, which is only useful when the distribution is close to a Gaussian.

Considering marginal coverage $C$, we observe that all methods, except for MCD, provide values close to the theoretical 90% coverage, showing similar capabilities and some degree of conservativeness. This is especially true for the conformalized methods (i.e., CQR and MCCP) for the smallest datasets (Boston housing and Concrete). However, MCD does not account for aleatoric uncertainty and, thus, the intervals generated from the quantiles of its predictive distribution are highly non-conservative. For larger datasets, the miscalibration of the conformalized methods becomes smaller and more stable as the number of observations to calibrate the prediction intervals increases. MCNF outperforms its MCD counterpart, highlighting the benefits of providing post hoc corrections over the latter. In addition, MCNF is computationally more efficient than MCD, especially when the number of samples to approximate the predictive distribution increases.

**Interval size.** Although all methods show some sensitivity to the actual uncertainty associated with the given dataset (as shown by the variability of the interval sizes $\tilde{\Delta}$ in Table 1), MCNF yields the smallest interval sizes while maintaining the expected 90% marginal coverage. While MCD also has small interval sizes, its marginal coverage is much worse than the other methods, failing to meet the expected threshold. Likewise, CQR and MCCP, in order to achieve the 90% through conformalization, yield conservative intervals which are reflected to their corresponding interval sizes. Accordingly, MCNF provides the best tradeoff between coverage and interval size, showing smaller interval sizes for similar marginal coverages.

**Complex distribution.** Unlike the multivariate datasets examined so far (e.g., Boston housing, Concrete), the Romano-Original and Romano-Mod univariate synthetic datasets allow for a clear visual comparison in reconstructing the data distribution. Figure 2 shows the prediction intervals from MCNF, MCD, and MCCP, alongside the predictive distributions across the predictor variable $x$. The

Table 1: Experimental results by dataset and metric over 20 runs. Methods are in columns; each cell shows mean $\pm$ std of coverage ($C$), median MAE ($\widetilde{\text{MAE}}$), and median prediction interval size ($\tilde{\Delta}$).

| Data | Metric | CQR | DQR | MCCP | MCD | MCQR | MCNF | NF |
|---|---|---|---|---|---|---|---|---|
| Boston H. | $C$ | 0.957±0.044 | 0.950±0.031 | 0.961±0.036 | 0.726±0.059 | 0.949±0.032 | 0.904±0.043 | 0.782±0.062 |
| | $\widetilde{\text{MAE}}$ | 0.454±0.248 | 0.315±0.040 | 0.439±0.196 | 0.078±0.009 | 0.318±0.040 | 0.078±0.009 | 0.073±0.009 |
| | $\tilde{\Delta}$ | 0.820±0.481 | 0.543±0.034 | 0.777±0.404 | 0.254±0.023 | 0.541±0.032 | 0.409±0.038 | 0.277±0.034 |
| Concrete | $C$ | 0.938±0.039 | 0.952±0.012 | 0.942±0.041 | 0.601±0.048 | 0.949±0.014 | 0.920±0.021 | 0.814±0.038 |
| | $\widetilde{\text{MAE}}$ | 0.355±0.044 | 0.372±0.039 | 0.375±0.038 | 0.113±0.012 | 0.378±0.038 | 0.085±0.012 | 0.084±0.008 |
| | $\tilde{\Delta}$ | 0.660±0.078 | 0.689±0.033 | 0.682±0.102 | 0.290±0.016 | 0.688±0.034 | 0.491±0.031 | 0.366±0.026 |
| Abalone | $C$ | 0.904±0.025 | 0.915±0.014 | 0.898±0.030 | 0.341±0.027 | 0.914±0.014 | 0.886±0.017 | 0.874±0.023 |
| | $\widetilde{\text{MAE}}$ | 0.344±0.036 | 0.350±0.031 | 0.344±0.038 | 0.100±0.005 | 0.354±0.031 | 0.099±0.007 | 0.098±0.005 |
| | $\tilde{\Delta}$ | 0.574±0.047 | 0.586±0.026 | 0.569±0.048 | 0.132±0.007 | 0.587±0.025 | 0.514±0.020 | 0.507±0.050 |
| Protein | $C$ | 0.899±0.009 | 0.922±0.009 | 0.901±0.010 | 0.354±0.018 | 0.921±0.009 | 0.927±0.006 | 0.887±0.018 |
| | $\widetilde{\text{MAE}}$ | 0.935±0.041 | 0.952±0.038 | 0.937±0.042 | 0.245±0.010 | 0.952±0.039 | 0.202±0.009 | 0.186±0.008 |
| | $\tilde{\Delta}$ | 1.827±0.022 | 1.861±0.019 | 1.828±0.019 | 0.406±0.011 | 1.857±0.019 | 1.781±0.033 | 1.673±0.062 |
| Wave | $C$ | 0.898±0.009 | 0.962±0.008 | 0.900±0.008 | 0.828±0.017 | 0.964±0.006 | 0.938±0.030 | 0.807±0.149 |
| | $\widetilde{\text{MAE}}$ | 0.005±0.001 | 0.008±0.001 | 0.005±0.001 | 0.001±0.0003 | 0.008±0.001 | 0.002±0.001 | 0.002±0.001 |
| | $\tilde{\Delta}$ | 0.010±0.001 | 0.016±0.002 | 0.010±0.001 | 0.009±0.0003 | 0.016±0.002 | 0.013±0.001 | 0.006±0.001 |
| Super | $C$ | 0.899±0.012 | 0.934±0.006 | 0.902±0.014 | 0.482±0.019 | 0.934±0.007 | 0.912±0.009 | 0.866±0.011 |
| | $\widetilde{\text{MAE}}$ | 0.283±0.026 | 0.299±0.025 | 0.289±0.024 | 0.110±0.006 | 0.303±0.024 | 0.103±0.005 | 0.096±0.004 |
| | $\tilde{\Delta}$ | 0.847±0.032 | 0.879±0.028 | 0.857±0.028 | 0.270±0.014 | 0.888±0.026 | 0.791±0.035 | 0.679±0.044 |
| R-OG | $C$ | 0.915±0.024 | 0.900±0.019 | 0.911±0.030 | 0.239±0.035 | 0.901±0.020 | 0.926±0.014 | 0.912±0.019 |
| | $\widetilde{\text{MAE}}$ | 2.208±0.150 | 2.176±0.134 | 2.200±0.152 | 0.511±0.103 | 2.182±0.134 | 0.406±0.206 | 0.496±0.154 |
| | $\tilde{\Delta}$ | 3.631±0.207 | 3.567±0.159 | 3.604±0.181 | 0.348±0.037 | 3.573±0.159 | 3.321±0.232 | 3.371±0.234 |
| R-MOD | $C$ | 0.912±0.028 | 0.948±0.027 | 0.918±0.032 | 0.540±0.0513 | 0.966±0.015 | 0.952±0.015 | 0.876±0.039 |
| | $\widetilde{\text{MAE}}$ | 0.201±0.029 | 0.234±0.039 | 0.223±0.032 | 0.074±0.008 | 0.274±0.036 | 0.067±0.006 | 0.060±0.012 |
| | $\tilde{\Delta}$ | 0.424±0.059 | 0.495±0.029 | 0.401±0.035 | 0.138±0.011 | 0.505±0.029 | 0.438±0.026 | 0.382±0.031 |
| Solubility | $C$ | 0.922±0.078 | 0.860±0.046 | 0.947±0.047 | 0.537±0.038 | 0.910±0.025 | 0.891±0.045 | 0.831±0.044 |
| | $\widetilde{\text{MAE}}$ | 0.749±0.289 | 0.490±0.115 | 0.836±0.185 | 0.221±0.019 | 0.637±0.108 | 0.207±0.014 | 0.215±0.024 |
| | $\tilde{\Delta}$ | 1.700±0.601 | 1.155±0.108 | 1.685±0.345 | 0.504±0.038 | 1.289±0.108 | 1.184±0.154 | 1.021±0.132 |

Romano-Mod dataset exhibits heteroskedasticity and varying distributions of the predicted variable for different $x$ values. MCNF effectively captures the multimodality of the predicted variable $y$ for small $x$, transitioning to a unimodal distribution as $x$ increases. This result is further corroborated by the overall better results achieved by MCNF against CQR, MCCP and DQR across the marginal coverage $C$, interval size $\tilde{\Delta}$ and MAE in both Romano-Original and Romano-Mod datasets.

**Predictive Model Impact.** Since MCNF is a post hoc UQ method, we evaluated the impact of the base predictive model on MCNF's performance. Thus, we assessed the performance using a well-trained predictive model and an underfitted model on the Concrete, Superconductivity, and Protein datasets. Selecting the well-trained predictive model entailed training 15 predictive models of the same architecture, for 100 epochs each, and selecting the model with the lowest RMSE. Similarly, selecting the underfitted model entailed training 15 predictive models for 6 epochs and selecting the model with the highest RMSE. Figure 3 shows the coverage and confidence interval for these two experiments, with results for the well-trained predictive model presented with a darker color and results for the underfitted model shown with a lighter shade color. Although the marginal coverage between the two predictive models and across most UQ methods (except from MCD) are similar and around the 90% threshold, the confidence interval plots (bottom) indicate that a well-trained predictive model provides, expectedly, narrower interval sizes. More importantly, though, even with an underfitted predictive model, MCNF yields narrower interval sizes than the state-of-the-art UQ methods and the coverage values are comparable to the well-trained predictive model. Accordingly, these results demonstrate that MCNF can adapt its UQ estimate based on the quality of the predictive model.

**Epistemic uncertainty propagation.** The performance metrics for MCNF without propagating the epistemic uncertainty through prediction errors via MCD sampling are shown in the last column of Table 1 (reported as NF). To produce these results, the MCNF workflow remains the same but prediction errors are not resampled. It is noteworthy that with this setup, there is a general

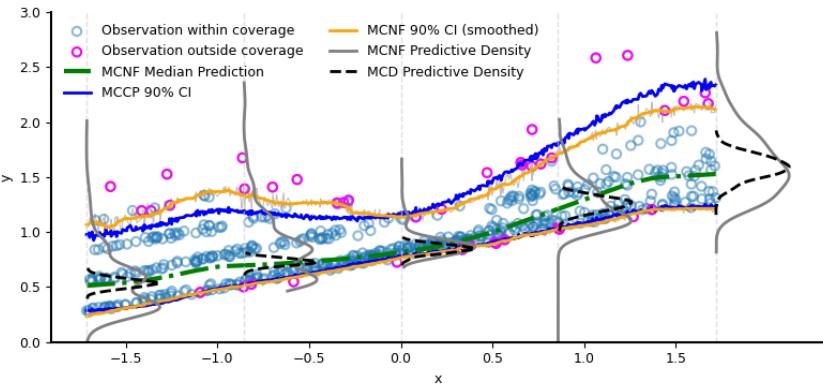

Figure 2: MCNF predictions (y) against the synthetic Romano-Mod dataset, generated as described in the Appendix. Blue circles represent data observations within the 90% marginal coverage of MCNF, whereas the pink circles fall outside this range. The orange interval delineates the MCNF smoothed marginal coverage (superimposed over the unsmoothed interval, in gray). The broken green line represents the median marginal coverage. The ridge lines represent kernel density estimates of the predictive distributions for the MCNF samples (gray) and the prior MCD samples (black).

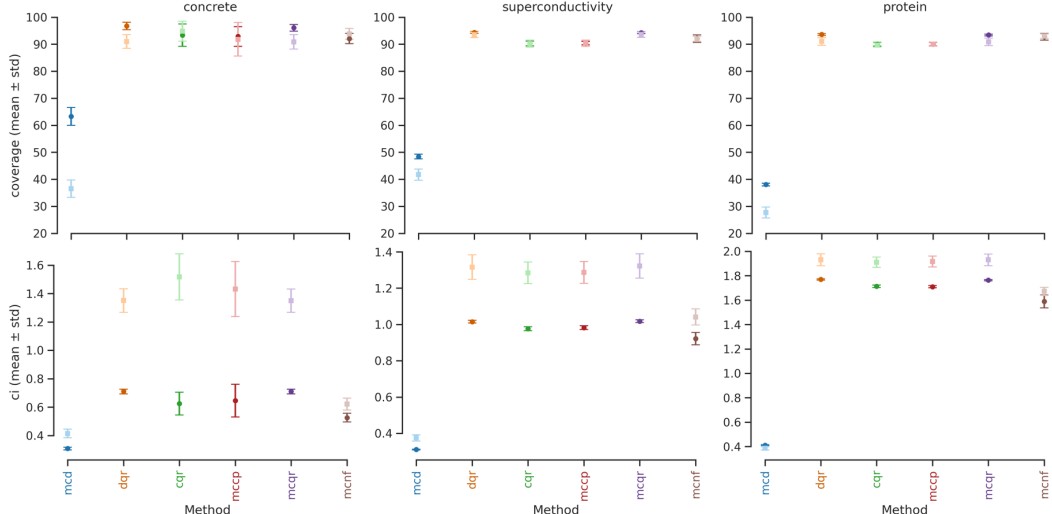

Figure 3: (best viewed in color) Coverage and confidence interval results for a well-trained predictive model (dark colored) and an underfitted predictive model (light colored).

performance decay for across all datasets included in the evaluation. When this source of uncertainty is not explicitly propagated, the prediction intervals deduced from the predictive distribution are consistently narrower than those obtained with the full MCNF.

## 5.2 MCNF application on Graph Neural Networks

We demonstrate the generalizability of MCNF to other deep learning model architectures through its application on a Graph Neural Network (GNN) predictive model [32] to make structure-based predictions of the physicochemical properties of molecules. The GNN model features three graph convolution layers based on the message passing mechanism, followed by an average readout layer to combine node-level encodings into a graph-level encoding. The next layers are a batch normalization layer to center and scale graph-level encoding, and a sequence of two dense layers of 300 neurons each (first a linear layer, then a second applying ReLU). Dropout was applied globally at a rate of 0.15. We use the Solubility dataset [33] that includes solubility data of 829 drug-like molecules.

We challenge MCNF by making it retrieve context for the NF component using an internal representation of a pre-trained GNN. Therefore, the parameters of the GNN remain constant while fitting the NF component of MCNF. The results for this experiment are shown at the bottom of Table 1. The obtained results are in agreement with those observed for the other datasets (Boston housing, Concrete, etc.) using deep regression models. In particular, all methods exhibit good performance in terms of marginal coverage, except for MCD. Among the well-calibrated methods, MCNF stands out by providing the best trade-off between coverage $C$, prediction interval sizes $\tilde{\Delta}$ and MAE.

## 6 Conclusion and Future Work

MCNF is a post hoc method that quantifies uncertainty in deep regression models by estimating the predictive distribution of the predicted variable. To achieve this, MCNF utilizes pre-trained deep regression models with dropout layers and models prediction errors using a shallow normalizing flow to correct prior MCD estimates. Through a comprehensive experimental evaluation comprising diverse datasets and state-of-the-art UQ methods, we demonstrate that prediction intervals from MCNF are well-calibrated, with smaller median sizes, providing richer information than baseline methods. The approach generalizes well to pre-trained GNNs, showing good calibration and adaptivity. Additionally, we mitigate the negative impact of outliers on the forward Kullback-Leibler divergence loss function by using MCD before weighing observations in a mini-batch. Future work could involve extending the MCNF formalism to classification problems and further characterizing the knowledge arising from the predictive model and its underlying assumptions. Furthermore, we will investigate techniques to improve the computational efficiency of MCNF by reducing the number of required MCD samples and coupling it with more efficient alternatives for building the prior knowledge.

## Acknowledgments and Disclosure of Funding

This work has been supported by the projects QCircle (grant agreement No 101059999) and SO-PRANO, GuardAI and AI4Work (grant agreements No 101120990, 101168067 and 101135990, respectively), funded by the European Union. Views and opinions expressed are however those of the author(s) only and do not necessarily reflect those of the European Union. Neither the European Union nor the granting authority can be held responsible for them.

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

## A MCNF Training

**Mini-batch strategy**

---

**Algorithm 2** Mini-batch building scheme to train the Normalizing Flow head of MCNF

---

**Inputs**: $\mathcal{D} = \{\mathbf{x}_n, y_n\}_{n=1}^N \{y_{i,\text{MCD}}\}_{i=1}^{n_{\text{MCD}}}$
**Parameters**: $\mathfrak{b} \equiv$ batch size
**Output**: $\mathfrak{d}$
1: Initialize $\mathfrak{d} = \{\}$, and $\mathcal{J} = \{1, ..., N\}$
2: **while** $|\mathcal{J}| > 0$ **do**
3:     Initialize $\mathfrak{d}_i = \{\}$
4:     **for** $k \leftarrow 1$ **to** $\mathfrak{b}$ **do**
5:         Generate a random index, $n \sim \mathcal{U}(1, |\mathcal{J}|)$ and pick $(\mathbf{x}_n, y_n)$
6:         Generate a random index $i \sim \mathcal{U}(1, n_{\text{MCD}})$ and draw $y_{n,i,\text{MCD}}$
7:         Calculate prediction error $\delta_n = y_n - y_{n,i,\text{MCD}}$
8:         Aggregate n-th observation to current mini-batch, $\mathfrak{d}_i = \mathfrak{d}_i \cup \{(\mathbf{x}_n, \delta_n)\}$
9:         Update indices set, $\mathcal{J} = \mathcal{J} \setminus \{n\}$
10:     Update mini-batches set, $\mathfrak{d} = \mathfrak{d} \cup \mathfrak{d}_i$

---

During MCNF training, in order to effectively propagate epistemic uncertainty to $p_{\boldsymbol{\theta},\boldsymbol{\psi}}(y|\mathbf{x}, \mathcal{D})$ when the latter is approximated as in Equation (6), we bootstrap the prediction errors. To this end, each mini-batch used to evaluate the gradients of the module based on the NFs is obtained by bootstrapping on a set of $n_{\text{MCD}}$ samples, $\{y_{i,\text{MCD}}\}_{i=1}^{n_{\text{MCD}}}$, previously generated from the distribution $p(y_{\text{MCD}}|\mathbf{x})$. Thus, the original training dataset $\mathcal{D} = \{\mathbf{x}_n, y_n\}_{n=1}^N$ is further processed at each iteration of the training steps to recalculate the prediction error $\delta_n$, as indicated in Algorithm 2. Consequently, the implemented mini-batch construction strategy enables every mini-batch to have a different realization of the prediction error per observation at each iteration.

## B Synthetic dataset

**Romano-Mod dataset.** A univariate stochastic process is introduced, by adapting the equation provided in Appendix B in [14]. The distribution that characterizes the uncertainty incorporates heteroskedasticity, which is a particularly relevant validation case to test whether the method adapts correctly to the local distribution of the data. The updated stochastic process is given:

$$y = \text{Poisson}\left(\sin x + \Delta\right) + (\beta\epsilon_1 + b) \cdot x + \delta(u \leq \bar{u}) \cdot \gamma \cdot \epsilon_2 \tag{9}$$

where $\delta()$ represents the Delta Kronecker function and $\epsilon_1$ and $\epsilon_2$ follow a standard Gaussian distribution. The parameters of the model are $\beta$ and $\Delta$, which condition the heteroskedasticity of the uncertainty distribution, as well as $\gamma$ and $\bar{u}$, which control the magnitude and rate of outliers in the sample generated. We introduce $b$ to induce a linear correlation between the predictor $x$ and the predicted variable $y$. Table 2 summarizes the values used for these parameters and Figure 4 illustrates the data generated.

Table 2: Parameters for the creation of the synthetic dataset.

| Data | Parameters | | | |
|------|---|---|---|---|
| | $\Delta$ | $\beta$ | $b$ | $\bar{u}$ |
| Romano-Original | 0.1 | 0.05 | 0.0 | 0.0 |
| Romano-Mod | 0.1 | 0.05 | 2.0 | 0.0 |

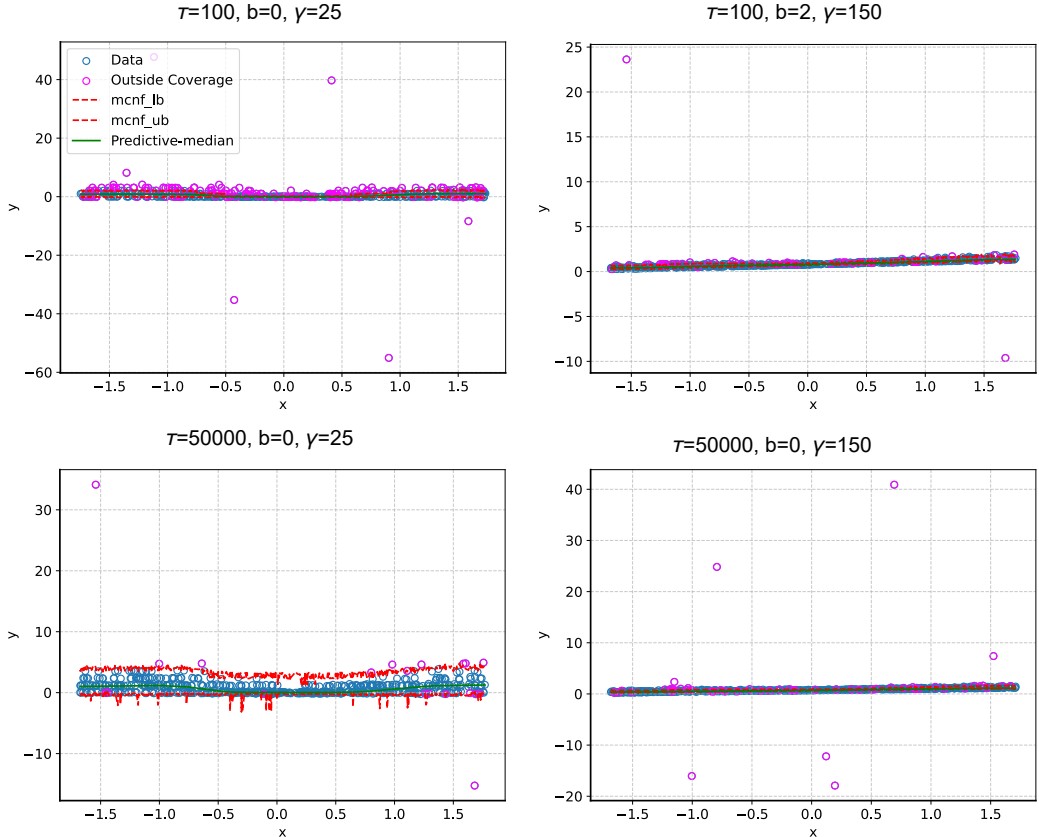

Figure 4: Visualisation of the synthetic data produced based on Equation (9).

## C   MCNF performance details

### C.1   Adaptivity of prediction intervals

In terms of prediction interval adaptivity, all examined UQ methods, including MCNF, are sensitive to the uncertainty inherent in the data, as evidenced by the variability of the quantiles of the interval sizes shown in Table 3. MCD has the narrowest intervals but covers a much smaller proportion of the true values, achieving a very poor coverage overall. Considering the UQ methods that perform well and converge to the expected marginal coverage (90%), MCNF provides the narrowest intervals and maintains similar marginal coverage to the other methods. Quantile MAE ($\text{MAE}_q$), where $\text{MAE}_q(X, Y) = \frac{1}{N} \sum_{i=1}^{N} |y_i - q_{0.05}(x_i)| + |y_i - q_{0.95}(x_i)|$, which provides additional evidence of the improved trade-off between coverage and interval sizes for MCNF, exhibits smaller values for similar marginal coverages.

### C.2   Ablation study

#### C.2.1   Hyperparameter study

We also examined the performance of MCNF for the following hyperparameters: number of epochs (epochs $\in \{20, 50, 100, 150\}$), number of normalizing flow samples ($n_{\text{NF}} \in \{100, 200, 500\}$), and number of Monte-Carlo Dropout samples ($n_{\text{MCD}} \in \{50, 100, 150\}$). For this evaluation, we employ the same prediction model across all tests, which was selected based on the lowest RMSE value, as described in Section 5.1.

Initially, we evaluate the number of epochs specified for training MCNF, examining epochs $\in \{20, 50, 100, 150\}$, as shown in Figure 5, whilst keeping the number of normalizing flow samples and Monte-Carlos Dropout samples fixed to $n_{\text{NF}} = 500$ and $n_{\text{MCD}} = 50$, respectively. Figure 6 illustrates

Table 3: Prediction interval sizes $\tilde{\Delta}_v(X,Y)$ for $v \in \{0.05, 0.25, 0.5, 0.75, 0.95\}$, where $\tilde{\Delta}_v(X,Y) = Q^n_{i=1}(q_{0.95}(x_i) - q_{0.05}(x_i))[v]$ and Quantile MAE, where $\mathrm{MAE}_q(X,Y) = \frac{1}{N}\sum_{i=1}^N |y_i - q_{0.05}(x_i)| + |y_i - q_{0.95}(x_i)|)$ for all UQ methods and datasets.

| Data | Metric | CQR | DQR | MCCP | MCD | MCQR | MCNF | NF |
|---|---|---|---|---|---|---|---|---|
| Boston H. | $\tilde{\Delta}_{0.05}$ | 0.650 ± 0.481 | 0.374 ± 0.031 | 0.605 ± 0.409 | 0.172 ± 0.018 | 0.373 ± 0.030 | 0.300 ± 0.027 | 0.163±0.019 |
| | $\tilde{\Delta}_{0.25}$ | 0.730 ± 0.482 | 0.458 ± 0.030 | 0.690 ± 0.407 | 0.212 ± 0.018 | 0.456 ± 0.029 | 0.352 ± 0.033 | 0.218±0.030 |
| | $\tilde{\Delta}_{0.5}$ | 0.820 ± 0.481 | 0.543 ± 0.034 | 0.777 ± 0.404 | 0.254 ± 0.023 | 0.541 ± 0.032 | 0.409 ± 0.038 | 0.277±0.034 |
| | $\tilde{\Delta}_{0.75}$ | 0.971 ± 0.494 | 0.695 ± 0.058 | 0.932 ± 0.400 | 0.318 ± 0.029 | 0.692 ± 0.057 | 0.507 ± 0.059 | 0.363±0.047 |
| | $\tilde{\Delta}_{0.95}$ | 1.315 ± 0.520 | 1.048 ± 0.084 | 1.269 ± 0.401 | 0.497 ± 0.054 | 1.044 ± 0.081 | 0.834 ± 0.092 | 0.617±0.132 |
| | $\mathrm{MAE}_q$ | 0.824 ±0.482 | 0.551 ±0.038 | 0.785 ±0.402 | 0.279 ±0.023 | 0.549 ±0.037 | 0.428 ±0.037 | 0.295±0.029 |
| Concrete | $\tilde{\Delta}_{0.05}$ | 0.430 ± 0.081 | 0.461 ± 0.019 | 0.458 ± 0.105 | 0.183 ± 0.015 | 0.463 ± 0.019 | 0.298 ± 0.027 | 0.193±0.024 |
| | $\tilde{\Delta}_{0.25}$ | 0.542 ± 0.075 | 0.574 ± 0.030 | 0.569 ± 0.097 | 0.238 ± 0.014 | 0.573 ± 0.030 | 0.394 ± 0.025 | 0.283±0.024 |
| | $\tilde{\Delta}_{0.5}$ | 0.660 ± 0.078 | 0.689 ± 0.033 | 0.682 ± 0.102 | 0.292 ± 0.016 | 0.688 ± 0.034 | 0.491 ± 0.031 | 0.366±0.026 |
| | $\tilde{\Delta}_{0.75}$ | 0.832 ± 0.099 | 0.859 ± 0.052 | 0.852 ± 0.115 | 0.382 ± 0.028 | 0.858 ± 0.052 | 0.621 ± 0.053 | 0.469±0.027 |
| | $\tilde{\Delta}_{0.95}$ | 1.045 ± 0.103 | 1.076 ± 0.068 | 1.072 ± 0.113 | 0.528 ± 0.049 | 1.072 ± 0.066 | 0.818 ± 0.091 | 0.676±0.068 |
| | $\mathrm{MAE}_q$ | 0.636 ±0.079 | 0.667 ±0.040 | 0.663 ±0.102 | 0.329 ±0.026 | 0.666 ±0.039 | 0.472 ±0.032 | 0.352±0.020 |
| Abalone | $\tilde{\Delta}_{0.05}$ | 0.357 ± 0.044 | 0.367 ± 0.020 | 0.349 ± 0.045 | 0.077 ± 0.007 | 0.370 ± 0.020 | 0.294 ± 0.017 | 0.277±0.030 |
| | $\tilde{\Delta}_{0.25}$ | 0.446 ± 0.046 | 0.457 ± 0.025 | 0.439 ± 0.048 | 0.109 ± 0.007 | 0.459 ± 0.025 | 0.380 ± 0.019 | 0.371±0.039 |
| | $\tilde{\Delta}_{0.5}$ | 0.574 ± 0.047 | 0.586 ± 0.026 | 0.569 ± 0.048 | 0.132 ± 0.007 | 0.587 ± 0.025 | 0.514 ± 0.020 | 0.507±0.050 |
| | $\tilde{\Delta}_{0.75}$ | 0.748 ± 0.043 | 0.757 ± 0.027 | 0.739 ± 0.048 | 0.163 ± 0.007 | 0.758 ± 0.027 | 0.744 ± 0.040 | 0.724±0.045 |
| | $\tilde{\Delta}_{0.95}$ | 1.117 ± 0.037 | 1.128 ± 0.038 | 1.109 ± 0.046 | 0.249 ± 0.013 | 1.127 ± 0.040 | 1.098 ± 0.052 | 1.092±0.046 |
| | $\mathrm{MAE}_q$ | 0.551 ±0.043 | 0.562 ±0.021 | 0.544 ±0.045 | 0.222 ±0.008 | 0.563 ±0.021 | 0.499 ±0.017 | 0.494±0.039 |
| Protein | $\tilde{\Delta}_{0.05}$ | 0.709 ± 0.080 | 0.744 ± 0.078 | 0.720 ± 0.081 | 0.113 ± 0.015 | 0.749 ± 0.080 | 0.341 ± 0.046 | 0.270±0.048 |
| | $\tilde{\Delta}_{0.25}$ | 1.549 ± 0.035 | 1.585 ± 0.026 | 1.557 ± 0.032 | 0.227 ± 0.019 | 1.584 ± 0.026 | 1.244 ± 0.111 | 1.155±0.152 |
| | $\tilde{\Delta}_{0.5}$ | 1.827 ± 0.022 | 1.861 ± 0.019 | 1.828 ± 0.019 | 0.406 ± 0.011 | 1.857 ± 0.019 | 1.781 ± 0.033 | 1.673±0.062 |
| | $\tilde{\Delta}_{0.75}$ | 2.010 ± 0.023 | 2.044 ± 0.022 | 2.009 ± 0.020 | 0.524 ± 0.009 | 2.038 ± 0.022 | 2.061 ± 0.026 | 1.933±0.044 |
| | $\tilde{\Delta}_{0.95}$ | 2.249 ± 0.032 | 2.284 ± 0.036 | 2.250 ± 0.032 | 0.762 ± 0.015 | 2.279 ± 0.038 | 2.393 ± 0.043 | 2.230±0.043 |
| | $\mathrm{MAE}_q$ | 1.349 ±0.048 | 1.383 ±0.041 | 1.354 ±0.047 | 0.576 ±0.014 | 1.382 ±0.041 | 1.172 ±0.059 | 1.048±0.088 |
| Wave | $\tilde{\Delta}_{0.05}$ | 0.004 ± 0.001 | 0.010 ± 0.001 | 0.004 ± 0.001 | 0.005 ± 0.000 | 0.010 ± 0.001 | 0.008 ± 0.001 | 0.003±0.001 |
| | $\tilde{\Delta}_{0.25}$ | 0.007 ± 0.001 | 0.013 ± 0.002 | 0.007 ± 0.001 | 0.007 ± 0.000 | 0.013 ± 0.002 | 0.010 ± 0.001 | 0.004±0.001 |
| | $\tilde{\Delta}_{0.5}$ | 0.010 ± 0.001 | 0.016 ± 0.002 | 0.010 ± 0.001 | 0.009 ± 0.000 | 0.016 ± 0.002 | 0.013 ± 0.001 | 0.006±0.001 |
| | $\tilde{\Delta}_{0.75}$ | 0.020 ± 0.001 | 0.026 ± 0.002 | 0.020 ± 0.001 | 0.012 ± 0.000 | 0.026 ± 0.002 | 0.022 ± 0.002 | 0.016±0.002 |
| | $\tilde{\Delta}_{0.95}$ | 0.040 ± 0.002 | 0.046 ± 0.003 | 0.041 ± 0.002 | 0.021 ± 0.001 | 0.047 ± 0.003 | 0.043 ± 0.004 | 0.039±0.005 |
| | $\mathrm{MAE}_q$ | 0.009 ±0.001 | 0.015 ±0.002 | 0.010 ±0.001 | 0.009 ±0.0002 | 0.016 ±0.002 | 0.013 ±0.001 | 0.006±0.001 |
| Super | $\tilde{\Delta}_{0.05}$ | 0.163 ± 0.015 | 0.196 ± 0.012 | 0.165 ± 0.014 | 0.058 ± 0.007 | 0.196 ± 0.012 | 0.141 ± 0.014 | 0.096±0.008 |
| | $\tilde{\Delta}_{0.25}$ | 0.286 ± 0.015 | 0.318 ± 0.012 | 0.289 ± 0.015 | 0.111 ± 0.006 | 0.319 ± 0.011 | 0.270 ± 0.024 | 0.220±0.012 |
| | $\tilde{\Delta}_{0.5}$ | 0.847 ± 0.032 | 0.879 ± 0.028 | 0.857 ± 0.026 | 0.270 ± 0.014 | 0.888 ± 0.026 | 0.791 ± 0.035 | 0.679±0.044 |
| | $\tilde{\Delta}_{0.75}$ | 1.769 ± 0.058 | 1.802 ± 0.061 | 1.768 ± 0.059 | 0.479 ± 0.020 | 1.799 ± 0.061 | 1.471 ± 0.089 | 1.336±0.078 |
| | $\tilde{\Delta}_{0.95}$ | 2.074 ± 0.060 | 2.105 ± 0.063 | 2.066 ± 0.059 | 0.735 ± 0.031 | 2.097 ± 0.064 | 2.087 ± 0.106 | 1.988±0.071 |
| | $\mathrm{MAE}_q$ | 0.610 ±0.025 | 0.641 ±0.025 | 0.612 ±0.029 | 0.292 ±0.007 | 0.642 ±0.024 | 0.566 ±0.023 | 0.462±0.022 |
| R-OG | $\tilde{\Delta}_{0.05}$ | 1.764 ± 0.181 | 1.698 ± 0.145 | 1.751 ± 0.207 | 0.110 ± 0.040 | 1.708 ± 0.146 | 1.832 ± 0.345 | 1.846±0.269 |
| | $\tilde{\Delta}_{0.25}$ | 2.374 ± 0.225 | 2.309 ± 0.200 | 2.361 ± 0.222 | 0.138 ± 0.046 | 2.316 ± 0.196 | 2.852 ± 0.193 | 2.785±0.239 |
| | $\tilde{\Delta}_{0.5}$ | 3.631 ± 0.207 | 3.567 ± 0.159 | 3.604 ± 0.181 | 0.348 ± 0.037 | 3.573 ± 0.159 | 3.321 ± 0.232 | 3.371±0.234 |
| | $\tilde{\Delta}_{0.75}$ | 4.262 ± 0.245 | 4.198 ± 0.225 | 4.249 ± 0.226 | 0.458 ± 0.035 | 4.209 ± 0.223 | 4.211 ± 0.091 | 4.093±0.124 |
| | $\tilde{\Delta}_{0.95}$ | 4.516 ± 0.297 | 4.446 ± 0.276 | 4.505 ± 0.281 | 0.557 ± 0.037 | 4.463 ± 0.276 | 4.373 ± 0.098 | 4.296±0.129 |
| | $\mathrm{MAE}_q$ | 2.615 ± 0.350 | 2.583 ± 0.350 | 2.597 ± 0.242 | 1.117 ± 0.166 | 2.589 ± 0.350 | 2.977 ± 0.395 | 2.886±0.254 |
| R-MOD | $\tilde{\Delta}_{0.05}$ | 0.329 ± 0.043 | 0.340 ± 0.025 | 0.319 ± 0.041 | 0.079 ± 0.007 | 0.339 ± 0.025 | 0.230 ± 0.016 | 0.188±0.016 |
| | $\tilde{\Delta}_{0.25}$ | 0.382 ± 0.042 | 0.392 ± 0.023 | 0.372 ± 0.041 | 0.105 ± 0.007 | 0.391 ± 0.022 | 0.300 ± 0.025 | 0.235±0.036 |
| | $\tilde{\Delta}_{0.5}$ | 0.491 ± 0.042 | 0.495 ± 0.029 | 0.483 ± 0.042 | 0.141 ± 0.006 | 0.494 ± 0.029 | 0.438 ± 0.026 | 0.382±0.031 |
| | $\tilde{\Delta}_{0.75}$ | 0.543 ± 0.045 | 0.543 ± 0.031 | 0.535 ± 0.042 | 0.168 ± 0.007 | 0.543 ± 0.030 | 0.511 ± 0.025 | 0.458±0.034 |
| | $\tilde{\Delta}_{0.95}$ | 0.661 ± 0.044 | 0.660 ± 0.036 | 0.656 ± 0.044 | 0.201 ± 0.009 | 0.660 ± 0.038 | 0.599 ± 0.036 | 0.528±0.048 |
| | $\mathrm{MAE}_q$ | 0.364 ± 0.064 | 0.433 ± 0.028 | 0.332 ± 0.037 | 0.187 ± 0.012 | 0.434 ± 0.029 | 0.350 ± 0.032 | 0.288±0.034 |
| Solubility | $\tilde{\Delta}_{0.05}$ | 1.157 ± 0.633 | 0.610 ± 0.076 | 1.153 ± 0.369 | 0.266 ± 0.021 | 0.756 ± 0.081 | 0.614 ± 0.084 | 0.446±0.075 |
| | $\tilde{\Delta}_{0.25}$ | 1.423 ± 0.611 | 0.867 ± 0.083 | 1.416 ± 0.365 | 0.380 ± 0.033 | 1.030 ± 0.086 | 0.933 ± 0.130 | 0.765±0.122 |
| | $\tilde{\Delta}_{0.5}$ | 1.700 ± 0.601 | 1.155 ± 0.108 | 1.685 ± 0.345 | 0.504 ± 0.039 | 1.289 ± 0.108 | 1.184 ± 0.154 | 1.021±0.132 |
| | $\tilde{\Delta}_{0.75}$ | 2.055 ± 0.582 | 1.511 ± 0.142 | 2.020 ± 0.338 | 0.618 ± 0.044 | 1.616 ± 0.139 | 1.500 ± 0.164 | 1.293±0.160 |
| | $\tilde{\Delta}_{0.95}$ | 2.693 ± 0.600 | 2.156 ± 0.214 | 2.603 ± 0.373 | 0.749 ± 0.043 | 2.220 ± 0.206 | 1.972 ± 0.152 | 1.723±0.255 |
| | $\mathrm{MAE}_q$ | 1.665 ± 0.590 | 1.124 ± 0.087 | 1.636 ± 0.335 | 0.597 ± 0.031 | 1.242 ± 0.085 | 1.119 ± 0.118 | 0.951±0.114 |

the results of evaluating the different normalizing flow sample values $n_{\mathrm{NF}} \in \{100, 200, 500\}$, setting epochs = 100 and $n_{\mathrm{MCD}} = 50$. Lastly, Figure 7 visualizes the ablation results for the different Monte-Carlo Dropout sample values $n_{\mathrm{MCD}} \in \{50, 100, 500\}$, whilst training the MCNF for epochs = 100 and sampling $n_{\mathrm{NF}} = 500$ normalizing flow samples. Our results demonstrate that all tested $n_{\mathrm{MCD}}$

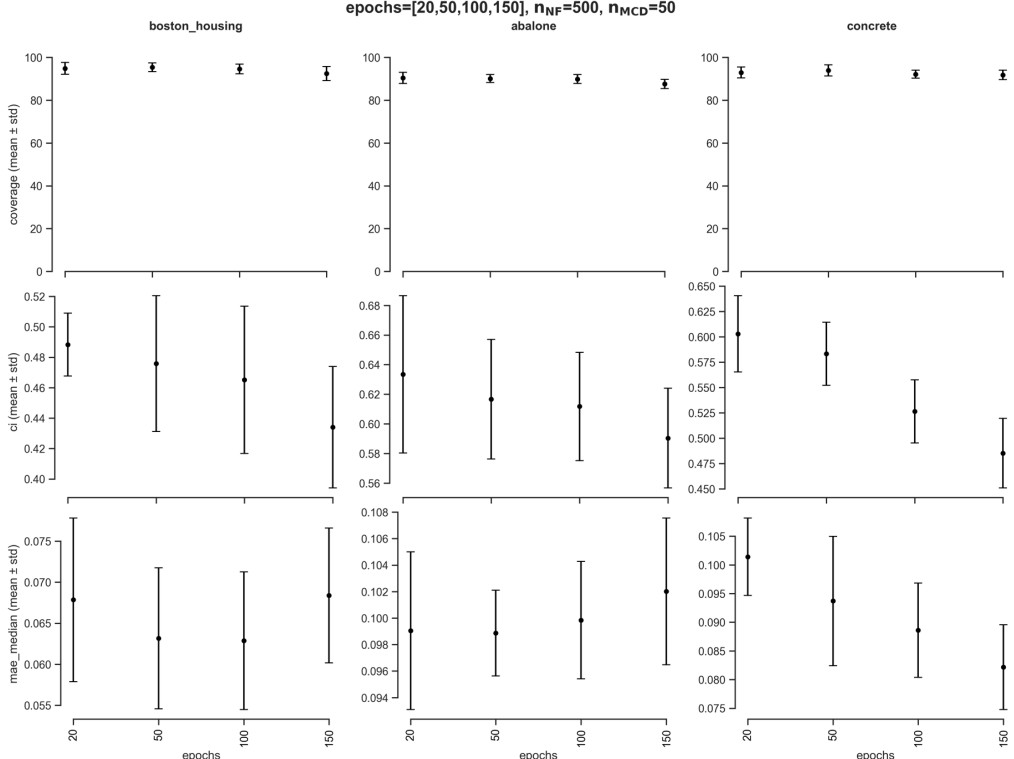

Figure 5: Evaluation of MCNF on Boston Housing, Abalone, and Concrete datasets for different epoch values (20, 50, 100, and 150).

values perform similarly in terms of the coverage metrics, and the difference in the confidence interval and the median MAE is negligible. Similar observations have been derived for the various epoch and $n_{NF}$ values evaluated.

### C.2.2 Outliers handling: testing impact of $\tau$ hyperparameter

As outlined in recent UQ survey papers [6], UQ methods can be impacted by outliers. Thus, we investigated the performance and response of MCNF to outlier data. Table 4 shows experiments carried out for different $\tau$, $b$ and $\gamma$ value combinations on MCNF coverage ($C$), median MAE ($\widetilde{\text{MAE}}$), and Quantile MAE ($\text{MAE}_q$). We observe that MCNF was sensitive to outliers when the effect size was close to zero, which likely results from biasing of the maximum likelihood estimator. The effect size relates to the degree of correlation between the feature variables and the response variable. We sought to mitigate this impact, given that low effect sizes (as well as sporadic large outlier values) frequently characterize data sets in many disciplines.

To test the impact of different $\tau$ values in Equation (8), with respect to changing effect size, we synthesize data (Section B) by modulating the slope ($b$) of the governing generative process to generate multiple sets of simulated data (Figure 8), where $b \in \{0, 0.1, 0.2, 0.3, 1, 2\}$. As the scale of the outliers relative to the rest of the data also changed with the slope, we modulated the $\gamma$ parameter to keep the outliers in the same approximate range. The outliers proportion was 1%. For each level of slope, we predicted from the MCNF method using the following $\tau$ values $\tau \in \{10^2, 2 \cdot 10^2, 3 \cdot 10^2, 10^3, 2 \cdot 10^3, 5 \cdot 10^3, 10^4, 10^5, 10^6\}$. The data distributions are shown in Figure 8.

Generally, the marginal coverage increases with the value of $\tau$ as the effect of the temperature scaling assigns equal importance to outliers and non-outliers. However, this comes at the cost of overly conservative prediction intervals and less adaptivity to the uncertainty in the data. No single value for $\tau$ can be considered universally optimal, as shown in Figure 8. This hyperparameter should be fine-tuned, potentially using a calibration set. When the ratio of the outlier magnitude over the effect size becomes smaller (larger values of the slope in this case) the importance of the former on the regular maximum likelihood procedure decreases, and values of $\tau$ can be smaller.

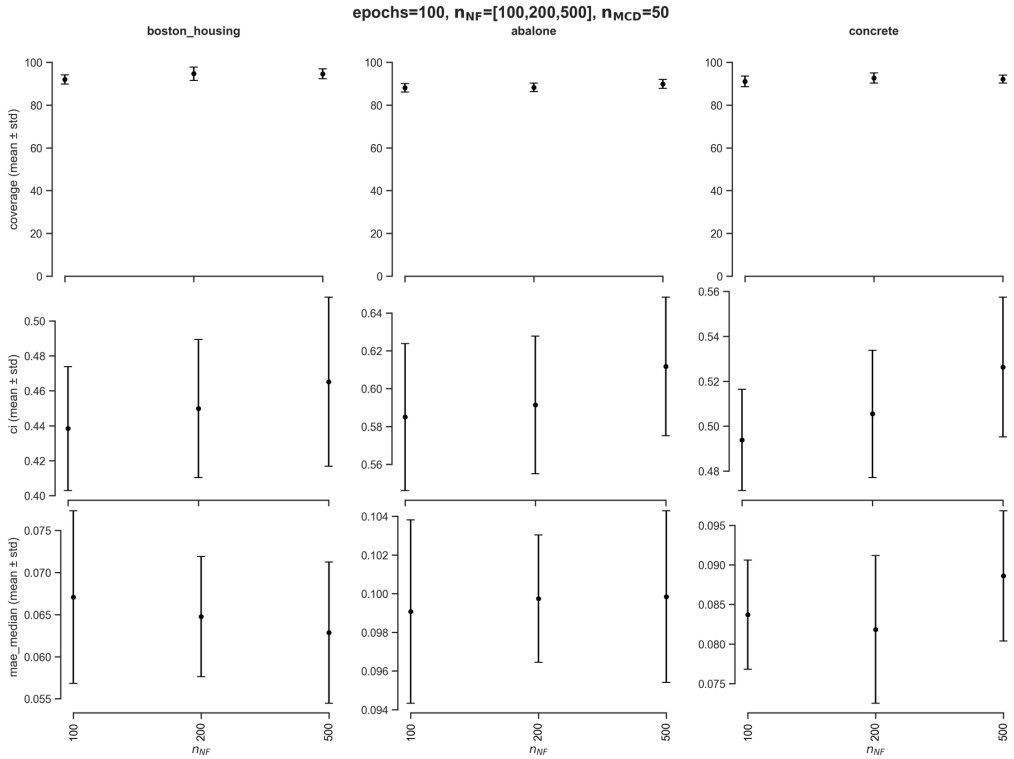

Figure 6: Evaluation of MCNF on Boston Housing, Abalone, and Concrete datasets for different $n_{NF}$ values (100, 200, and 500).

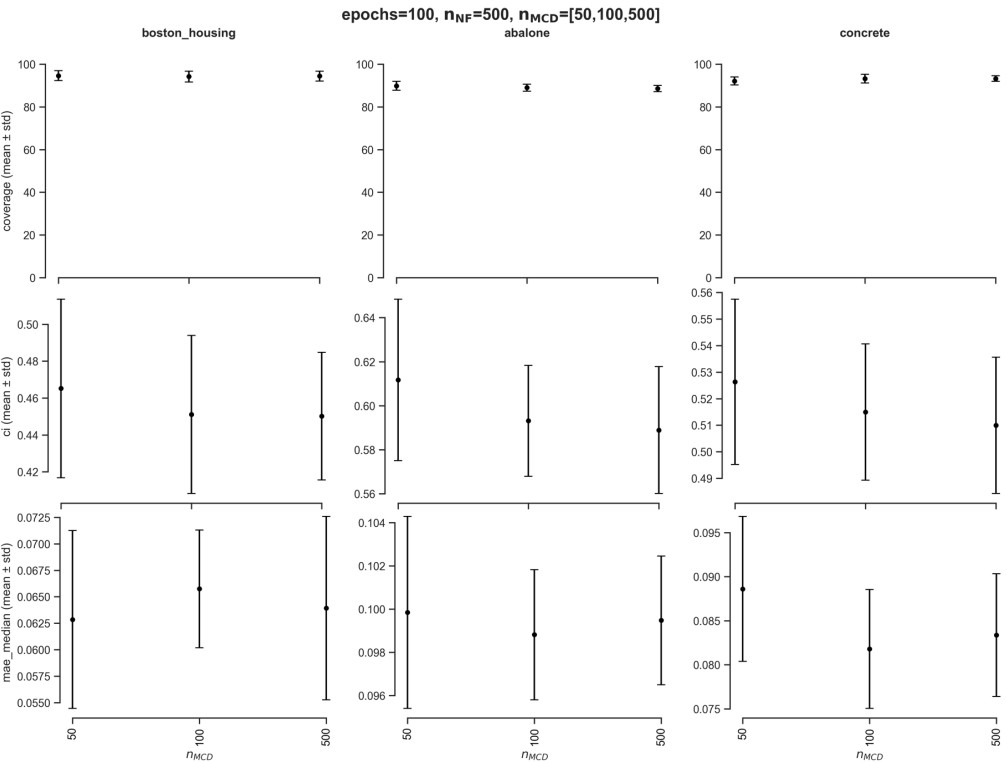

Figure 7: Evaluation of MCNF on Boston Housing, Abalone, and Concrete datasets for different $n_{MCD}$ values (50, 100, and 500).

Table 4: Evaluation of the impact of outliers for different $\tau$, $b$, and $\gamma$ value combinations on MCNF coverage ($C$), median MAE ($\widetilde{\text{MAE}}$), and Quantile MAE ($\text{MAE}_q$).

| $\tau$ | $b$ | $\gamma$ | $C$ | $\widetilde{\text{MAE}}$ | $\text{MAE}_q$ |
|---|---|---|---|---|---|
| 100 | 0 | 25 | 0.8100 | 0.2061 | 1.3039 |
| 100 | 0.1 | 25 | 0.7883 | 0.1845 | 0.9799 |
| 100 | 0.2 | 25 | 0.7900 | 0.2307 | 0.9087 |
| 100 | 0.3 | 25 | 0.8200 | 0.1271 | 0.7115 |
| 100 | 1 | 100 | 0.8917 | 0.1040 | 0.4217 |
| 100 | 2 | 150 | 0.9117 | 0.2258 | 0.0474 |
| 200 | 0 | 25 | 0.8067 | 0.2887 | 1.4055 |
| 200 | 0.1 | 25 | 0.8467 | 0.3229 | 1.2045 |
| 200 | 0.2 | 25 | 0.8567 | 0.1541 | 0.9818 |
| 200 | 0.3 | 25 | 0.8800 | 0.1594 | 0.8330 |
| 200 | 1 | 100 | 0.9350 | 0.1284 | 0.7240 |
| 200 | 2 | 150 | 0.9333 | 0.0625 | 0.2758 |
| 300 | 0 | 25 | 0.8467 | 0.2001 | 1.3449 |
| 300 | 0.1 | 25 | 0.8467 | 0.3229 | 1.2045 |
| 300 | 0.2 | 25 | 0.8783 | 0.1607 | 1.3490 |
| 300 | 0.3 | 25 | 0.8800 | 0.1594 | 0.8330 |
| 300 | 1 | 100 | 0.9283 | 0.1070 | 0.4160 |
| 300 | 2 | 150 | 0.9283 | 0.0570 | 0.2793 |
| 1000 | 0 | 25 | 0.9067 | 0.6610 | 1.9665 |
| 1000 | 0.1 | 25 | 0.8867 | 0.1307 | 1.1857 |
| 1000 | 0.2 | 25 | 0.9017 | 0.1413 | 0.9706 |
| 1000 | 0.3 | 25 | 0.9133 | 0.1875 | 0.8647 |
| 1000 | 1 | 100 | 0.9283 | 0.0978 | 0.3980 |
| 1000 | 2 | 150 | 0.9367 | 0.0640 | 0.2923 |
| 2000 | 0 | 25 | 0.9000 | 0.2752 | 1.6500 |
| 2000 | 0.1 | 25 | 0.9400 | 0.1788 | 1.4421 |
| 2000 | 0.2 | 25 | 0.9233 | 0.1430 | 1.1068 |
| 2000 | 0.3 | 25 | 0.9183 | 0.1163 | 0.8374 |
| 2000 | 1 | 100 | 0.9350 | 0.0807 | 0.4679 |
| 2000 | 2 | 150 | 0.9467 | 0.0576 | 0.2600 |
| 5000 | 0 | 25 | 0.9083 | 0.8049 | 1.6046 |
| 5000 | 0.1 | 25 | 0.9467 | 0.1651 | 1.2981 |
| 5000 | 0.2 | 25 | 0.9300 | 0.2350 | 1.1888 |
| 5000 | 0.3 | 25 | 0.9183 | 0.2916 | 0.7965 |
| 5000 | 1 | 100 | 0.9200 | 0.0957 | 0.4036 |
| 5000 | 2 | 150 | 0.9500 | 0.0439 | 0.3584 |
| 10000 | 0 | 25 | 0.9200 | 0.3249 | 3.4255 |
| 10000 | 0.1 | 25 | 0.9467 | 0.1651 | 1.2981 |
| 10000 | 0.2 | 25 | 0.9133 | 0.1299 | 1.0917 |
| 10000 | 0.3 | 25 | 0.9300 | 0.1082 | 0.8284 |
| 10000 | 1 | 100 | 0.9417 | 0.1113 | 0.5516 |
| 10000 | 2 | 150 | 0.9400 | 0.0632 | 0.2897 |
| 50000 | 0 | 25 | 0.9800 | 0.2216 | 6.4337 |
| 50000 | 0.1 | 25 | 0.9450 | 0.1666 | 1.7998 |
| 50000 | 0.2 | 25 | 0.9667 | 0.1667 | 1.8143 |
| 50000 | 0.3 | 25 | 0.9433 | 0.1299 | 1.2884 |
| 50000 | 1 | 100 | 0.9417 | 0.1113 | 0.5516 |
| 50000 | 2 | 150 | 0.9400 | 0.0599 | 0.3046 |
| 100000 | 0 | 25 | 0.9867 | 0.3689 | 7.2465 |
| 100000 | 0.1 | 25 | 0.9867 | 0.2644 | 6.5211 |
| 100000 | 0.2 | 25 | 0.9800 | 0.2321 | 2.9301 |
| 100000 | 0.3 | 25 | 0.9650 | 0.1731 | 1.8576 |
| 100000 | 1 | 100 | 0.9500 | 0.1085 | 0.5747 |
| 100000 | 2 | 150 | 0.9833 | 0.0654 | 0.6146 |

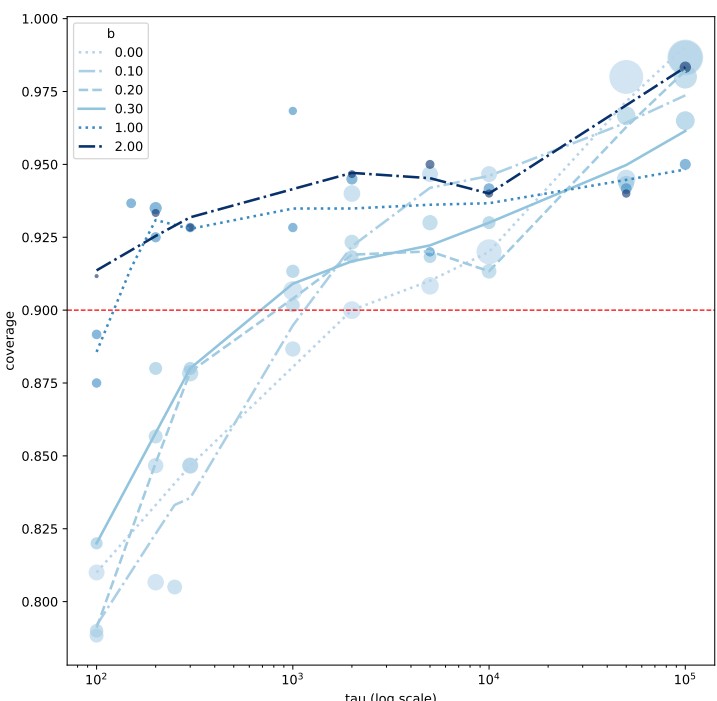

Figure 8: MCNF marginal coverage with respect to $\tau$ for different slope $b$ values.

