# OpenReview forum: "Uncertainty Quantification for Deep Regression using Contextualised Normalizing Flows"
_NeurIPS.cc/2025/Conference — NeurIPS 2025 poster_

### Official Review · Reviewer_onw7 · 2025-06-24

**Clarity:** 3
**Significance:** 2
**Originality:** 2
**Rating:** 4
**Confidence:** 2

**Summary:**

The Authors propose an uncertainty quantification technique for regression models based on neural networks. This technique makes use of Montecarlo dropout to get a prior estimate of the epistemic uncertainty of the model, and combines it post-hoc with normalizing flows to get a distribution of the prediction error (includung aleatoric uncertainty).

As reflected in my confidence score, my evaluation capabilities for this paper are limited as UQ is not my main area of research.

**Questions:**

- In the experimental evaluation, you report averaged results over 20 independent runs. Initialization and the stochastic gradient dynamics are a source of epistemic uncertainty. Do you have estimates on the magnitude of this uncertainty compared to the one introduced by MCD? Does this question make sense at all?

**Ethical Concerns:**

["NO or VERY MINOR ethics concerns only"]

**Final Justification:**

I am satisfied with the rebuttal, and I will keep my positive score.

**Limitations:**

Not discussed

**Quality:**

3

**Strengths And Weaknesses:**

### Strengths
- Thorough experimental evaluation addressing multiple metrics and datasets. Among the experimental evaluations, I also appreciated a lot the study on the impact of the pre-trained predictive model.
- The paper is generally well written, although many variables are introduced with proper explanation (see weaknesses).

#### Weaknesses
- Many variables are introduced without proper explanation. Among these there are: $\mathcal{D}$ on line 97, $\Xi, \phi$ on line 132, $\theta, \psi$ on line 124
- Lacking details on the computational complexity of the method. The inference task, in particular, requires first performing Monte Carlo dropout sampling, and then a pass on the normalizing flow model.

---

> ### Author Rebuttal · Authors · 2025-07-31
>
> Dear reviewer onw7,
>
> Thank you very much for your valuable review and your questions! Please find answers to your questions below:
>
> > Many variables are introduced without proper explanation. Among these there are: $\mathcal{D}$ on line 97, $\Xi, \phi$ on line 132, $\theta, \psi$ on line 124
>
> A: We appreciate the careful reading of the manuscript by the reviewer and the highlighting of some examples of undefined symbols. Below, symbols missing an appropriate description are defined:
>
> * $\mathcal{D}$: is the set of observations $\{\mathbf{x}_n, y_n\}_{n=1}^{N_{|\mathcal{D|}} available for training.
>
> * $\theta$, $\psi$: these denote de parameters for the series of transformations making up the Normalizing Flow and the parameters for the base Gaussian distribution characterizing the NF latent space, respectively
>
> * $\Xi$: this symbol denotes the parameters of the predictive model, $\phi(\mathbf{x};\Xi)$.
>
> We have gone through the paper and found an additional erratum in line 230 when defining the median prediction interval width:
>
> Current form:
>
> * $\tilde{\Delta}(X,Y) = median_{i=1}^n (q_{1-\alpha}(x_i),q_{\alpha}(x_i))$
>
> Fixed form:
>
> * $\tilde{\Delta}(X,Y) = median_{i=1}^n (q_{1-\alpha}(x_i) - q_{\alpha}(x_i))$
>
> All these changes will be added to the camera ready version of the manuscript, if accepted.
>
>
> > Lacking details on the computational complexity of the method. The inference task, in particular, requires first performing Monte Carlo dropout sampling, and then a pass on the normalizing flow model.
>
> A: The computational complexity at inference time incurred by MCNF is linear with respect to n_MCD +n_NF. More specifically, this involves running n_MCD forward passes to build the context vector as per Equation 5 and then drawing n_NF samples to produce the prediction output y and its density.
>
> We have also investigated the robustness of MCNF using n_NF ∈ {100, 200, 500} and n_MCD ∈ {50, 100, 150} for the Boston Housing, Abalone, and Concrete datasets in Appendix C.2.1. Our results demonstrate that MCNF performs similarly in terms of the coverage metrics, and the difference in the confidence interval and the median MAE is negligible for all n_MCD values. Similar observations have been derived for the various n_NF values evaluated, with small differences observed in terms of slightly higher coverage as n_NF increases at the cost of slightly higher confidence intervals. These results demonstrate the overall robustness of MCNF with respect to its n_MCD and n_NF hyperparameters, indicating that MCNF adopters can comfortably find appropriate values for their domains and datasets.
>
> We analysed the computational complexity of MCNF and the other UQ methods to assess the memory and computational time  consumed for the evaluated datasets.  Our findings indicate that MCNF runs significantly faster than MCCP, and in some datasets (abalone, concrete) similar to baselines CQR and DQR.
>
>
>
> > In the experimental evaluation, you report averaged results over 20 independent runs. Initialization and the stochastic gradient dynamics are a source of epistemic uncertainty. Do you have estimates on the magnitude of this uncertainty compared to the one introduced by MCD? Does this question make sense at all?
>
> A: Thank you for this interesting question, which definitely makes sense. As reported in the Evaluation-Performance Metrics section, each independent run includes sources of epistemic uncertainty unrelated to the number of parameter (i.e. models) sets of the base predictive model that could explain the data, which is captured by MCD. This is primarily due to the resulting data partition model and the effect of stochastic gradient dynamics.  Then, the 20 independent runs average out the impact of the sources of epistemic uncertainty.
>
> In Figure 3, we compare the performance of MCNF using a weak and a strong base predictive model, thus focusing primarily on the effect of stochastic gradient dynamics due to MCNF training. These results show the impact that the quality of the base predictive model has on MCNF and the corresponding epistemic uncertainty arising from MCD, which is smaller than the corresponding impact in the other methods we evaluated (e.g., MCD, MCCP, CQR).
>
> Performing a fine-grained analysis to further disentangle the uncertainty is definitely an interesting direction for future work in order to establish further the MCNF capabilities in working with base predictive models and datasets with diverse characteristics.

---

> > ### Comment · Reviewer_onw7 · 2025-08-05
> >
> > I thank the Authors for their responses and clarifications. I am satisfied with the rebuttal, and I will keep my positive score.

---

### Official Review · Reviewer_eM1j · 2025-06-27

**Clarity:** 3
**Significance:** 3
**Originality:** 4
**Rating:** 4
**Confidence:** 4

**Summary:**

The paper presents a posthoc UQ method leveraging upon Monte Carlo Dropout (MCD) and normalizing flows.  Specifically, the normalizing flow model is trained to learn a distribution for the output variable $y$ given a feature vector derived somewhere in the base prediction model and summary statistics of the mean and log-variance of the output of the base model under MCD. The experiments somewhat demonstrate that the error quantiles are better calibrated of the new method against other state of the art UQ methods for regression problems.

**Questions:**

Can the authors explain why the interval length $\Delta$ matters and why it is not intrinsic with respect to the CDF of possible values of $y$?

What are the values for $n_{MCD}$ and $n_{NF}$ used in the paper?  Have the authors investigated the robustness of the MCNF with respect to these parameters?

In Figure 3, I get why the interval spread should be larger for the underfitted (less trained) model, and this is prominent in the strong baselines. However, the increase in the spread for MCNF is less prominent.  Why would this be desirable?  I would think that epistemic uncertainty increases in the underfitted models, and thus $\Delta$ should jump much higher.

**Ethical Concerns:**

["NO or VERY MINOR ethics concerns only"]

**Final Justification:**

After considering the author rebuttal and the other reviews, I happy with my recommendation of borderline accept.

The paper is an interesting post-training uncertainty quantification method based on MCD and NF. It is not of low computational complexity, but it is effective in capturing calibrated confidence intervals and exhibiting high accuracy. The concern of one reviewer about the use of the MCD baseline is reasonable. Overall, the paper has some novelty and mostly sufficient experiments with one flaw. Borderline accept seem appropriate. Personally, I am happy to see this paper included in the conference program if there is room.

**Limitations:**

The paper mentions the computational complexity as a possible limitation, especially for larger prediction models, in the concluding section. Also, the concluding section mentions adapting the method for classification as future work. The paper could discuss the computational complexity for inference in the experimental section with respect to the other UQ baseline methods.

**Paper Formatting Concerns:**

None!

**Quality:**

3

**Strengths And Weaknesses:**

The proposed UQ methods makes sense and seems effective for capturing arbitrary probability distributions that can be multimodal with complex structure. The essence of the idea is that one can learn the distribution of possible output from a context variable consisting of features from a base forward pass model along with statistics of the output of the base model under MCD. This idea is intriguing and appears novel.

The experiments seem to compare with a sufficient number of baseline state-of-the-art methods. I appreciate that the paper considers both DQR and GNNs as base prediction models.

I do have some concerns with the experiments. First, I am not clear why the interval length should matter.  The interval length is dictated by the unknown ground truth distribution of the output $y$ given knowledge of the input raw features $x$, and thus $\Delta$ may need to large or small in light of the spread of this ground truth distribution.

While it is great to measure the coverage for a desired coverage of 90%, it think it would be more useful to compute an expected absolute calibration error by considering the absolute difference of the coverage and the desired coverage of x% for x ranging from 0 to 100.

I like Figure 2 as a means to show the need to accommodate  complex multimodal distributions in UQ methods.

The paper does not discuss the computational cost for inference via sampling for MCNF relative to the computational cost of the baseline methods.

As a reader, it would be easier for me to understand $\log \sigma^2$ as the log-variance rather than $\log s^2$ as given in (5).

In (6), it is not clear what depends on the Monte Carlo index "$i$" for the summation. Somehow, it is hidden in $\delta$, which is only revealed in Algorithm 1.

---

> ### Author Rebuttal · Authors · 2025-07-31
>
> Dear reviewer eM1j,
>
> Thank you very much for your valuable review and your questions! Please find answers to your questions below:
>
> > As a reader, it would be easier for me to understand as the log-variance rather than as given in (5). In (6), it is not clear what depends on the Monte Carlo index " " for the summation. Somehow, it is hidden in, which is only revealed in Algorithm 1.
>
> A: Thank you for raising this notation-related point. With respect to the log-variance, we preferred $\log s^2$ over $\log \sigma^2$ to denote that the statistic used to build the context is the sample variance and not the true (population) variance. Regarding Equation 6, we agree with the reviewer’s observation and will fix it accordingly to make the index explicit inside the summation operator.
>
>
> > Can the authors explain why the interval length Δ matters and why it is not intrinsic with respect to the CDF of possible values of y ?
>
> A: As the reviewer suggests, $\Delta$ is intrinsic with respect to the underlying distribution of the predicted variable y. However, different distribution spreads may account for the same (or very) close marginal coverage. Therefore, for the same level of coverage level (or coverage miscalibration), narrower predictive distributions (MCD, MCNF) and intervals (for interval-based approaches, e.g. DQR, CQR or MCCP) are preferred. Additionally, the spread of predictive distributions and the width of the prediction intervals should adapt conveniently to the actual uncertainty (as discussed elsewhere, [2107.07511] A Gentle Introduction to Conformal Prediction and Distribution-Free Uncertainty Quantification), avoiding overly conservative assessments. Since the experiments carried out showed that MCNF coverage miscalibration is comparable to that of the selected baselines, $\Delta$ becomes a meaningful metric for comparing the performance of all the considered methods. We also discuss the adaptivity, showing the prediction interval width variability (using sample quantiles) in Table 3 in Appendix C.2.1.
>
>
> > What are the values for n_MCF and n_NF used in the paper? Have the authors investigated the robustness of the MCNF with respect to these parameters?
>
> A: We have indeed investigated the robustness of MCNF using n_NF ∈ {100, 200, 500} and n_MCD ∈ {50, 100, 150} for the Boston Housing, Abalone, and Concrete datasets in Appendix C.2.1. Our results demonstrate that MCNF perform similarly in terms of the coverage metrics, and the difference in the confidence interval and the median MAE is negligible for all n_MCD values. Similar observations have been derived for the various n_NF values evaluated, with small differences observed in terms of slightly higher coverage as n_NF increases at the cost of slightly higher confidence intervals. These results demonstrate the overall robustness of MCNF with respect to its n_MCD and n_NF hyperparameters, indicating that MCNF adopters can comfortably find appropriate values for their domains and datasets.
>
>
> > In Figure 3, I get why the interval spread should be larger for the underfitted (less trained) model, and this is prominent in the strong baselines. However, the increase in the spread for MCNF is less prominent. Why would this be desirable? I would think that epistemic uncertainty increases in the underfitted models, and thus should jump much higher.
>
> A: Epistemic uncertainty indeed increases in underfitted models and that reflects on the larger widths of the prediction intervals for all of the UQ methods, as evidenced by Figure 3. The key factor that explains differences in how much the interval width increases for MCNF as compared to the remaining baselines lies in the fact that the baselines account not only for the data-intrinsic aleatoric uncertainty and the increase in the epistemic uncertainty, but also for the predictive model (DNN / GNN) bias. Conformal prediction methods (CQR and MCCP) need to significantly correct the uncalibrated intervals to guarantee the design marginal coverage while accounting for this bias. DQR and MCQR also require much larger intervals for the underfitted model. In contrast, MCNF trains the NF head of the method over the predictions of the underfitted predictive model, which recenters the uncertainty distribution, thus decoupling and removing the prediction bias from the aleatoric and epistemic sources of uncertainty. The first doesn’t change while the second is still propagated as shown in Figure 3.
>
>
> > The paper mentions the computational complexity as a possible limitation, especially for larger prediction models, in the concluding section. Also, the concluding section mentions adapting the method for classification as future work. The paper could discuss the computational complexity for inference in the experimental section with respect to the other UQ baseline methods.
>
> A: The computational complexity at inference time incurred by MCNF is linear with respect to n_MCD +n_NF. More specifically, this involves running n_MCD forward passes to build the context vector as per Equation 5 and then drawing n_NF samples to produce the prediction output y and its density. As discussed above, MCNF is robust to the n_MCD and n_NF values. We will add these clarifications in the final version of our paper.

---

> > ### Comment · Reviewer_eM1j · 2025-08-05
> >
> > I thank the authors for their responses and clarifications.  I am happy with their responses.

---

### Official Review · Reviewer_fJpF · 2025-07-03

**Clarity:** 3
**Significance:** 3
**Originality:** 3
**Rating:** 4
**Confidence:** 3

**Summary:**

The paper proposes a normalizing-flow-based post-hoc uncertainty quantification method that works on top of an already trained monte-carlo dropout model. The prediction mean and variance (or rather log-variance) of the output of the MCD model along with certain hidden layers of the MCD model are used to to form the condition (or context) of the NF model. The NF model itself represents the density of the residual \delta around the mean of the base model prediction. The method is explained in detail, and then empirical evaluation is provided.

**Questions:**

- The original monte-carlo dropout method outputs an aleatoric mean and variance, and the dropout part is only used to quantify epistemic uncertainty (ie, uncertainty of the mean and variance parameters themselves). Is this considered in the design of the NF method? Is it considered during MCD evaluation as a baseline?
- The mcd samples are reduced to mean and variance before feeding them to NF model (mind you, if I understand correctly, this is the mean and variance of the \mu parameter of the gaussian aleatoric distribution predicted by MCD). This appears to lose any shape information that the base model might predict for the mean (and any information predicted by the model for the aleatoric variance). Can you comment on this?
- Is there a reason that no NLL evaluation is done on the datasets, as is standard? Similarly, is there a reason that the quasi-standard UCI regression uncertainty benchmark is not presented along with the evaluations currently in the paper? (As is done e.g. in the original monte-carlo dropout paper and a large number of papers since).
- In the beginning, e.g. eq 3, we are made to believe that the normalizing flow transforms the output distribution of the base model into the aleatoric posterior of y through monte-carlo sampling. Yet after a while it turns out that in fact the output distribution of the base model is summarized into mean/var as context for the NF, and the actual base-distribution being transformed is in fact a certain z_{n,k} (e.g. line 180), I assume a gaussian, as is usual in normalizing flows. Does this not mean that the initial description is quite misleading? Did I miss something here?

**Ethical Concerns:**

["NO or VERY MINOR ethics concerns only"]

**Final Justification:**

The authors managed to address most of my concerns in their rebuttal. However, I still believe that one of the baselines (MCD) is presented unfairly in the experiments section: if my understanding is correct, the constant term for aleatoric variance is not included in the calculation, severely handicapping the method. This is not a critical issue, as it is only one baseline of 3, but it does weaken the paper.

**Limitations:**

yes

**Paper Formatting Concerns:**

line 138 typo: sampled vs sample

**Quality:**

3

**Strengths And Weaknesses:**

Overall, this is a quite strong paper. Strengths:
- Innovative idea, providing a method for distribution-free uncertainty quantification
- Well structured paper, (mostly) clear descriptions
- Sufficient evaluation
- Proper source-code provided

Weaknesses:
- Standard evaluation not included (UCI Benchmark)
- Possible confusion around around MCD predictive distributions (see questions)
- Unclear theoretical predictive power w.r.t. being distribution free (see questions)
- Confusing framing of the base-distribution of the normalizing flow (see questions)

Weak reject for now, but I expect to increase the score if the questions are answered in a convincing way.

---

> ### Author Rebuttal · Authors · 2025-07-31
>
> Dear reviewer fJpF,
>
> Thank you very much for your valuable review and your questions! Please find answers to your questions below:
>
> > The original monte-carlo dropout method outputs an aleatoric mean and variance, and the dropout part is only used to quantify epistemic uncertainty (ie, uncertainty of the mean and variance parameters themselves). Is this considered in the design of the NF method? Is it considered during MCD evaluation as a baseline?
>
> A: Monte Carlo Dropout (MCD) approximates the epistemic uncertainty of a model's estimates of the predicted variable **y**. On top of this, the method optionally accepts augmenting the number of outputs of the predictive model to include estimates of the aleatoric mean and variance, in which case a Gaussian assumption of the aleatoric uncertainty is made. Therefore, MCD allows obtaining a distributional uncertainty over the Gaussian assumption for this specific implementation. However, MCD can also be applied to approximate the distribution of the epistemic uncertainty of the predicted variable **y** itself. This is the framework that we have considered in this work. For MCFN, no distributional assumption is made for the total uncertainty (epistemic + aleatoric). Epistemic uncertainty is introduced via MCD sampling, while aleatoric uncertainty is inherent to the data used for training. Both of them are propagated together through the prediction error, $\delta_{n,k} = y_{n} - y_{n,k,\text{MCD}}$, which is learnt by the Normalizing Flow (NF) head underpinning MCNF.
>
>
> > The mcd samples are reduced to mean and variance before feeding them to NF model (mind you, if I understand correctly, this is the mean and variance of the \mu parameter of the gaussian aleatoric distribution predicted by MCD). This appears to lose any shape information that the base model might predict for the mean (and any information predicted by the model for the aleatoric variance). Can you comment on this?
>
> A: Following the discussion on the previous question, the prediction error, $\delta_{n,k}$, propagates the actual distributional information of MCD applied to the base model (epistemic uncertainty) as well as that of the aleatoric uncertainty (intrinsic to the training data).
>
> MCD prior estimates are summarised as the reviewer states, but with the sole purpose of building the context vector for the NF head of the MCNF method. This contextualization aims at providing location and spread information of the prior distribution. During training, different realizations of the prediction error are passed to the model by drawing samples from the MCD distribution as described in Algorithm 2 in Appendix A, describing the MCNF training.
>
>
> > Is there a reason that no NLL evaluation is done on the datasets, as is standard? Similarly, is there a reason that the quasi-standard UCI regression uncertainty benchmark is not presented along with the evaluations currently in the paper? (As is done e.g. in the original monte-carlo dropout paper and a large number of papers since).
>
> A: Concerning showing the NLL values, we did not consider incorporating these as, in our opinion, NLL is not an easily interpretable quantity, as its value depends on the data underlying distribution. Consequently, its magnitude cannot be used to make comparisons across datasets. This becomes even more apparent for continuous distributions as the NLL has no lower bound since densities may be larger than one, leading to negative NLLs. Besides, we cannot report NLL for all of the baselines included in our comparison. Should the reviewer believe that adding this information would add further clarity to our paper, we can easily incorporate these values for all of the datasets.
>
> Concerning the experimental set used, we used the same datasets employed in the state-of-the-art methods evaluated in our papers, i.e., CQR [1], MCCP [2] and MCD [3]. More specifically,
>
> * the Boston Housing dataset is used in [2] and [3]
> * the Concrete dataset is used in [1-3]
> * the Concrete dataset is used in [1-3]
> * the Abalone dataset  is used in [3]
> * the Protein Structure dataset is used in [2] and [3]
> * the Wave and Superconductivity datasets are used in [2]
> * the Romano-Original is a synthetic dataset produced in [1]
>
> As reported in the Evaluation section - Benchmarks, the datasets mentioned above have been retrieved from the UCI machine learning repository (https://archive.ics.uci.edu)
>
> Furthermore, we extended the Romano-Original dataset that has univariate predictor samples and few large outliers with support for multimodal distributions. The distribution that characterizes the uncertainty incorporates heteroskedasticity, which is a particularly relevant validation case to test whether MCNF adapts correctly to the local distribution of the data.
>
> Also, we demonstrated the applicability and benefits of MCNF by comparing it against the other methods using a GNN as the predictive model and the solubility dataset [4] that includes solubility data of 829 drug-like molecules.
>
> Taken together, our empirical evaluation adopts the established practice in the area, using the benchmark datasets from the UCI repository and conforming to the approach adopted in [1-3].
>
> [1] Romano, Y., Patterson, E., & Candes, E. (2019). Conformalized quantile regression. Advances in neural information processing systems, 32.
>
> [2] Bethell, D., Gerasimou, S., & Calinescu, R. (2024, March). Robust uncertainty quantification using conformalised Monte Carlo prediction. In Proceedings of the AAAI Conference on Artificial Intelligence (Vol. 38, No. 19, pp. 20939-20948).
>
> [3] Gal, Y., & Ghahramani, Z. (2016, June). Dropout as a bayesian approximation: Representing model uncertainty in deep learning. In international conference on machine learning (pp. 1050-1059). PMLR.
>
> [4] Mario Lovri´c, Kristina Pavlovi´c, Petar Žuvela, Adrian Spataru, Bono Luˇci´c, Roman Kern, and Ming Wah Wong. Machine learning in prediction of intrinsic aqueous solubility of drug-like compounds: Generalization, complexity, or predictive ability? Journal of Chemometrics, 35(7-8), jul 2021
>
>
> > In the beginning, e.g. eq 3, we are made to believe that the normalizing flow transforms the output distribution of the base model into the aleatoric posterior of y through monte-carlo sampling. Yet after a while it turns out that in fact the output distribution of the base model is summarized into mean/var as context for the NF, and the actual base-distribution being transformed is in fact a certain z_{n,k} (e.g. line 180), I assume a gaussian, as is usual in normalizing flows. Does this not mean that the initial description is quite misleading? Did I miss something here?
>
> A: In MCNF, the MCD prior (which is a source of the epistemic uncertainty) is propagated through the prediction error $\delta$ and used to inform the NF head of the method about the location and spread of this epistemic uncertainty through the context, c, as argued in previous answers; therefore Equation 3 and, particularly, Equation 6 hold. On the other hand, z_{n,k} represents the latent representation of the NF, for which we chose a Gaussian distribution with trainable parameters. The key point is that $z_{n,k} \neq y_{n,k,\text{MCD}}$, entailing that the MCD prior should not be assumed to be the NF base distribution, which can be selected arbitrarily. We are sorry for the confusion that the writing may have caused and will clarify this aspect in the camera-ready version of our paper.
>
>
> > line 138 typo: sampled vs sample
>
> A: Thank you for carefully reviewing the manuscript and reporting this typo. We will correct it in the final version of our manuscript.

---

> ### Comment · Reviewer_fJpF · 2025-08-05
>
> Thank you for the thoughtful response!
>
> > A: Monte Carlo Dropout (MCD) approximates the epistemic uncertainty of a model's estimates of the predicted variable y. On top of this, the method optionally accepts augmenting the number of outputs of the predictive model to include estimates of the aleatoric mean and variance, in which case a Gaussian assumption of the aleatoric uncertainty is made. Therefore, MCD allows obtaining a distributional uncertainty over the Gaussian assumption for this specific implementation. However, MCD can also be applied to approximate the distribution of the epistemic uncertainty of the predicted variable y itself. This is the framework that we have considered in this work. For MCFN, no distributional assumption is made for the total uncertainty (epistemic + aleatoric). Epistemic uncertainty is introduced via MCD sampling, while aleatoric uncertainty is inherent to the data used for training. Both of them are propagated together through the prediction error, $\delta_{n,k} = y_{n} - y_{n,k,\text{MCD}}$, which is learnt by the Normalizing Flow (NF) head underpinning MCNF.
>
> Thank you for clarifying. I would suggest emphasizing this fact in the paper, as 1. MCD is not described mathematically in the paper, thus one is lead to believe that you use it in the form it was published in (i.e., modeling epistemic + gaussian aleatoric distribution). 2. The notation $y_{MCD}$ around eq 2-3 also suggest this: $y$ generally means the variable itself, not its expectation.
>
> You didn't answer whether you consider the aleatoric distribution when evaluating MCD as a baseline. Do you?
>
> > Should the reviewer believe that adding this information would add further clarity to our paper, we can easily incorporate these values for all of the datasets
>
> I accept the reasoning for not including NLL, so don't feel pressured to include it (any time spent on this could probabily used more effectively making improvements to the clarity of the text).
>
> However, to also motivate my question about it, I just want to say that NLL usually serves as the traditional performance metric that is used to compare probabilistic method performance, however flawed it may be. As such, it helps place any method in context of prior uncertainty quantification methods, e.g. [1-12], which are all measured on the same datasets (UCI benchmark), therefore NLL not being comparable across datasets is not an issue.
>
> [1] Louizos & Welling. (2016). Structured and efficient variational deep learning with matrix Gaussian posteriors
> [2] Springenberg, Klein, Falkner & Hutter. (2016). Bayesian optimization with robust Bayesian neural networks
> [3] Gal & Ghahramani. (2016). Dropout as a Bayesian approximation: Representing model uncertainty in deep learning
> [4] Sun, Chen & Carin. (2017). Learning structured weight uncertainty in Bayesian neural networks
> [5] Lakshminarayanan, Pritzel & Blundell. (2017)
> [6] Mukhoti, Stenetorp & Gal. (2018). On the importance of strong baselines in Bayesian deep learning
> [7] Ghosh, Yao & Doshi-Velez. (2019). Model selection in Bayesian neural networks via horseshoe priors
> [8] Amini, Schwarting, Soleimany & Rus. (2020). Deep evidential regression
> [9] Goulet, Nguyen & Amiri. (2021). Tractable approximate Gaussian inference for Bayesian neural networks
> [10] El-Laham, Dalmasso, Fons & Vyetrenko. (2023). Deep Gaussian mixture ensembles
> [11] Gawlikowski et al. (2023). A survey of uncertainty in deep neural networks
> [12] Deka, Nguyen & Goulet. (2024). Analytically tractable heteroscedastic uncertainty quantification in Bayesian neural networks for regression tasks
>
> > We are sorry for the confusion that the writing may have caused and will clarify this aspect in the camera-ready version of our paper
>
> Your explanation makes sense, I indeed got the wrong impression on the first reading. Thank you for the clarification!
>
> I acknowledge that this might be the result of my background, but to me, the paper would be *much more* digestable if it was explicitly framed as a way to model aleatoric uncertainty on top of another method (MCD) to handle epistemic uncertainty, which is my current understanding of the method. Further, this framing would much more naturally allow questions such as "how is the epistemic uncertainty of the shape of the aleatoric distribution handled by the method"? Which now I think is a limitation of the method, as it only considers epistemic uncertainty of the mean. (Although it's a a very small limitation, as epistemic uncertainty of shape parameters in my experience doesn't really matter in practice). Feel free to correct me on this though.

---

> > ### Author Response · Authors · 2025-08-05
> >
> > > Thank you for clarifying. I would suggest emphasizing this fact in the paper, as 1. MCD is not described mathematically in the paper, thus one is lead to believe that you use it in the form it was published in (i.e., modeling epistemic + gaussian aleatoric distribution). The notation $y_{MCD}$ around eq 2-3 also suggest this: $y$ generally means the variable itself, not its expectation.
> >
> > A: Thank you for this suggestion for making these clarifications, which we will implement in the camera ready version of the paper, if accepted.
> >
> >
> > > Q: You didn't answer whether you consider the aleatoric distribution when evaluating MCD as a baseline. Do you?
> >
> > A: We apologize if our previous answer was ambiguous regarding this matter. In brief, no, we did not consider the aleatoric source of the uncertainty when evaluating MCD as a baseline.
> >
> > This was not an arbitrary decision, though. To the best of our understanding, in [1506.02142] Dropout as a Bayesian Approximation: Representing Model Uncertainty in Deep Learning, MCD is proposed to describe the predictive posterior of y, i.e. $p(y|x,\mathcal{D})$. Samples from this distribution are drawn by taking several forward passes of the underlying predictive model. Then, predictive mean and variance are estimated by assessing the sample mean and variance from the set of estimates obtained with MCD (as outlined in Section 4 of that paper), which are approximations of the first two moments of $p(y|x,\mathcal{D})$. Because of this, our interpretation is that there is no explicit treatment of the aleatoric uncertainty in the MCD seminal paper.
> >
> > The strategy that the reviewer suggests is perfectly valid, yet an extension of the original MCD to account for the aleatoric uncertainty, which ultimately leads to the distributional uncertainty of the Gaussian hypothesis over the aleatoric uncertainty.
> >
> >
> > > I accept the reasoning for not including NLL, so don't feel pressured to include it However, to also motivate my question about it, I just want to say that NLL usually serves as the traditional performance metric that is used to compare probabilistic method performance, however flawed it may be.
> >
> > A: Thank you for these interesting points. Since we can extract the NLL values quite easily, we are more than happy to include this information in the camera-ready version of the paper.
> >
> >
> > > I acknowledge that this might be the result of my background, but to me, the paper would be much more digestable if it was explicitly framed as a way to model aleatoric uncertainty on top of another method (MCD) to handle epistemic uncertainty, which is my current understanding of the method.
> >
> > A: The reviewer’s interpretation of how MCNF operates is correct. The Normalizing Flow (NF) head enables a distribution-free estimation of the total uncertainty (as opposed to, for example, MCD combined with the Gaussian hypothesis). To do that, the NF learns the predictive model prediction errors, $\delta$, which are augmented by propagating the epistemic uncertainty modelled with MCD, $\delta_{n,k} = y_{n} - y_{n,k,\text{MCD}}$. This procedure allows informing the NF of both the aleatoric and epistemic uncertainties, and how the latter is conveyed to influence the shape of the former.
> >
> > To make the concept clearer and illustrate how MCNF builds on top of MCD to account for both sources of uncertainty, let’s consider a dummy example using a noisy sinusoid function.
> >
> >     def sample_dataset(start, end, n):
> >         x = np.linspace(start, end, n)
> >         sample_mean = [math.sin(i/2) for i in x]
> >         sample_var = [((abs(start)+abs(end))/2 - abs(i))/16 for i in x]
> >         y = stats.norm(sample_mean, sample_var).rvs()
> >         return x[:,None], y[:,None]
> >
> > Assume that the range of values of the input variable x in the training data is [-7.5, +7.5] (i.e., start=-7.5, end=7.5, n=100). Since the sinusoid function is noisy, i.e., there is aleatoric uncertainty, the MCD estimates of the predictive variable $y$ would fail to describe the total uncertainty. As we move out from the training range, MCD reflects the increase in the epistemic uncertainty providing wider prediction intervals. In contrast, MCNF corrects MCD by including the effects of the aleatoric uncertainty, while also exhibiting the modulated effects of epistemic uncertainty.
> >
> > (unfortunately, the NeurIPS rules forbid us from adding any images to our response that would provide clear evidence of how MCNF incorporates both sources of uncertainty.)

---

> > > ### Comment · Reviewer_fJpF · 2025-08-06
> > >
> > > > The strategy that the reviewer suggests is perfectly valid
> > >
> > > Thank you for the clarification and sorry for being a bit slow, yes indeed what you write makes perfect sense, I was the one who was confused.
> > >
> > > > Because of this, our interpretation is that there is no explicit treatment of the aleatoric uncertainty in the MCD seminal paper.
> > >
> > > However, I'm not sure this one is true - while they do indeed estimate the moments from the MCD samples, they also add a constant term ($\tau^{-1}I_D$, see unnumbered equations between eq 6 and 7 in the MCD paper), making the full predictive variance the sum of the sample variance and a constant term.This does makes the prediction homoscedasctic rather than heteroscedastic, but is also a representation of aleatoric uncertainty.
> > >
> > > (Also note that in the Gaussian formula, they only need to account for the constant term, as the epistemic variance is included via the monte-carlo integration, see eq8 of the original paper).
> > >
> > > So I guess my only remaining question is whether you account for this constant?

---

> > > > ### Author Response · Authors · 2025-08-07
> > > >
> > > > > So I guess my only remaining question is whether you account for this constant?
> > > >
> > > > A: We thank the reviewer for the clarification. We now understand the motivation for this question.
> > > >
> > > > Concerning the precision term of the total variance, we chose not to incorporate it into the MCNF formalism as the primary objective is to utilise MCD specifically to address and propagate epistemic uncertainty, while the NF head of MCNF effectively handles aleatoric uncertainty. Consequently, we excluded this term when integrating MCD into the baseline uncertainty quantification (UQ) methods because the original MCD definition imposes inherent restrictions, i.e., the imposed homoscedasticity and the fact that the aleatoric uncertainty is controlled by the weight decay λ and the prior length-scale l chosen for training. This implication suggests that not employing weight decay ($\lambda=0$), potentially a valid design choice, would result in infinite precision, thereby indicating the absence of aleatoric uncertainty. Similarly, the prior length-scale l is not learnt directly and is typically fixed, as reported in the original MCD paper experiments, with a standard value of $10^{-2}$.
> > > >
> > > > To adequately account for the actual precision, it is preferable to refer to the extended procedure proposed by Kendall and Gal in 2017 (https://dl.acm.org/doi/10.5555/3295222.3295309 - What uncertainties do we need in Bayesian deep learning for computer vision?). Hence, the predictive variance is learnt from the data along with the predictive mean, enabling heterocedasticity. Despite its premise, we did not consider this approach in our work, as it would necessitate modifications to the architecture and (partial) retraining of the predictive model, which does not align with the focus of this study on post-hoc uncertainty quantification methods.
> > > >
> > > > Finally, we performed a search of papers published in NeurIPS/ICML/AAAI to investigate whether the precision term τ is used when using MCD. Our findings, albeit not complete, indicate that many research papers opt not to adopt this constant term.
> > > >
> > > > * Simple and Scalable Predictive Uncertainty Estimation using Deep Ensembles, NeurIPS 2017, https://dl.acm.org/doi/10.5555/3295222.3295387
> > > > * Predictive uncertainty estimation via prior networks, NeurIPS 2018, https://dl.acm.org/doi/10.5555/3327757.3327808
> > > > * Beyond unimodal: generalising neural processes for multimodal uncertainty estimation, NeurIPS 2023, https://dl.acm.org/doi/10.5555/3666122.3667951
> > > > * Out of Distribution Data Detection Using Dropout Bayesian Neural Networks, AAAI 2022, https://cdn.aaai.org/ojs/20757/20757-13-24770-1-2-20220628.pdf
> > > > * Capturing Uncertainty in Unsupervised GPS Trajectory Segmentation Using Bayesian Deep Learning, AAAI 2021, https://cdn.aaai.org/ojs/16115/16115-13-19609-1-2-20210518.pdf
> > > >
> > > > We thank the reviewer for bringing this topic to our attention. We will make explicit how MCD was applied within MCNF.

---

> > > > > ### Comment · Reviewer_fJpF · 2025-08-07
> > > > >
> > > > > Whether you incorporate the constant into MCNF is your decision as the author of the method. However, when measuring MCD as a *baseline*, it is not correct to omit a crucial part of the method. Especially since, as discussed before, modeling aleatoric uncertainty is a main point of your approach.
> > > > >
> > > > > I admit that the original MCD paper/code are somewhat fuzzy about the choice for $\tau$: they indeed write that $\tau$ can be calculated based on $l$ and $\lambda$, but in practice they use bayesian optimization / grid search to find the optimal value for $\tau$ in the actual implementations (see e.g. [6]). Not sure if this is explained in the original paper itself and I don't have the time to investigate right now.
> > > > >
> > > > > However, omitting the constant aleatoric uncertainty term and then validating against data that does include aleatoric uncertainty is a huge misrepresentation of the performance of the baseline, as in my experience aleatoric variance is usually much larger than the epistemic variance in practice.
> > > > >
> > > > > [6] Mukhoti, Stenetorp & Gal. (2018). On the importance of strong baselines in Bayesian deep learning

---

> > > > > > ### Author Response · Authors · 2025-08-07
> > > > > >
> > > > > > Thank you for this thoughtful comment. We appreciate you highlighting the importance of faithfully implementing baselines.
> > > > > >
> > > > > > As mentioned, we followed the established practice in using MCD as reported in similar research papers published in NeurIPS/ICML/AAAI:
> > > > > >
> > > > > > * Simple and Scalable Predictive Uncertainty Estimation using Deep Ensembles, NeurIPS 2017, https://dl.acm.org/doi/10.5555/3295222.3295387
> > > > > > * Predictive uncertainty estimation via prior networks, NeurIPS 2018, https://dl.acm.org/doi/10.5555/3327757.3327808
> > > > > > * Robust uncertainty quantification using conformalised Monte Carlo prediction, AAAI 2024, https://dl.acm.org/doi/abs/10.1609/aaai.v38i19.30084
> > > > > >
> > > > > > which also make the same simplification.
> > > > > >
> > > > > > We agree that omitting the constant aleatoric uncertainty term in the MCD baseline could lead to an underestimation of its total predictive uncertainty, especially in settings where aleatoric uncertainty is significant.
> > > > > >
> > > > > > We believe, however, that the absence of the constant term does not necessarily diminish the strength or significance of our current results, especially since MCNF is compared against state-of-the-art UQ methods like CQR and MCCP. The empirical evaluation demonstrates that MCNF provides the best tradeoff between coverage and interval size, showing a smaller interval for similar marginal coverages.
> > > > > >
> > > > > > To improve the clarity and transparency of our experimental evaluation and enhance its reproducibility, we will report this MCD simplification explicitly in the paper.

---

> > > > > > > ### Author Response · Authors · 2025-08-07
> > > > > > >
> > > > > > > Addendum: We report in the paper (lines 243-244) that the MCD implementation we used does not consider aleatoric uncertainty
> > > > > > >
> > > > > > > "However, MCD does not account for aleatoric uncertainty and, thus, the intervals generated from the quantiles of its predictive distribution are highly non-conservative".
> > > > > > >
> > > > > > > Should the reviewer believe that adding this information elsewhere in the text (e.g., in the Comparative Methods paragraph) would add further clarity to our paper, we can easily incorporate it.

---

> > > > > > > > ### Comment · Reviewer_fJpF · 2025-08-08
> > > > > > > >
> > > > > > > > Thank you, must have skipped over that part when reading the text.
> > > > > > > >
> > > > > > > > As your rebuttal managed to address most of my concerns, I'm increasing my score, however I still can't recommend straigh up accept (5), as I believe that such usage of MCD as is presented in the experiments section is still a misrepresentation of the performance of the method. You could improve the text by further highlighting that the MCD version benchmarked is not the original version but a handicapped one, but even if the paper is honest about it, I don't really see a reasonable explanation for doing this, given the simplicity of the method.

---

### Official Review · Reviewer_8wsk · 2025-07-03

**Clarity:** 3
**Significance:** 2
**Originality:** 3
**Rating:** 4
**Confidence:** 4

**Summary:**

This paper introduces Monte Carlo Normalizing Flow (MCNF), a novel post-hoc method for uncertainty quantification (UQ) in deep regression models. The core idea is to improve upon the uncertainty estimates generated by Monte Carlo Dropout (MCD) by using them to "contextualize" a normalizing flow (NF). This NF then models the distribution of the prediction errors. The method aims to provide not just prediction intervals but the full predictive posterior distribution, which can capture complex features like asymmetry and multimodality.

**Questions:**

The paper mentions that using the first two moments is a reasonable approximation under a normality assumption for the MCD distribution. How does MCNF's performance change if this assumption is strongly violated, for instance in a clearly bimodal case?

The training loss includes a temperature-scaled softmax to weigh observations, mitigating the effect of outliers. In the experiments, τ is set to a very large value (1e10), effectively giving equal weight to all observations. Why was this choice made?

How does MCNF compare to other methods that also use normalizing flows for uncertainty quantification, but perhaps in a different manner ?

A strength of conformal prediction is that it enjoys coverage guarantees for the queried alpha. Does MCNF enjoy some theoretical properties (even under ideal assumptions) ?

**Ethical Concerns:**

["NO or VERY MINOR ethics concerns only"]

**Final Justification:**

I thank the authors for the detailed answers provided to my questions. Based on the additional empirical evidence and arguments they gave, I raised my score by one unit.

As the authors indicate that the paper is focused on post hoc uncertainty quantification, I recommend this position to appear in the very title of the paper.

**Limitations:**

Yes

**Paper Formatting Concerns:**

None.

**Quality:**

2

**Strengths And Weaknesses:**

pros:
- The proposed MCNF method presents a clever and novel way to combine existing techniques (MCD and NFs)
- The proposed approach can be deployed on already trained deep learning models with dropout layers without the high computational cost of retraining.

cons:
- The approach relies on MCD and the MCD context provided to the normalizing flow might be misleading
- The approach introduces additional computational steps at inference time (operations of the normalizing flow).
- The results are difficult to read as several (antagonistic) criteria are necessary to compare the methods. It is hard to know which method delivers the best trade-off

---

> ### Author Rebuttal · Authors · 2025-07-31
>
> Dear reviewer 8wsk,
>
> Thank you very much for your valuable review and your questions! Please find answers to your questions below:
>
> > The paper mentions that using the first two moments is a reasonable approximation under a normality assumption for the MCD distribution. How does MCNF's performance change if this assumption is strongly violated, for instance in a clearly bimodal case?
>
> A: In MCNF, the sample mean and log variance of the samples drawn from the MCD distribution are used to build the Normalizing Flow’s context vector only (Equation 5), ultimately affecting the assessment of the likelihood as given by Equation. 6. Due to the introduction of the prediction error $\delta_{n,k} = y_{n} - y_{,n,k\text{MCD}}$ for every n-th observation (input) and k-th estimate drawn from the MCD distribution, the Normalizing Flow (NF) model is learning from the actual MCD distribution which reflects the epistemic uncertainty. Accordingly, even if the ground truth was a completely deterministic variable (i.e., no aleatoric uncertainty), and the distribution of the epistemic uncertainty was bimodal, the NF would be aware of that bimodality by virtue of the samples drawn for the prediction error. Summarizing the MCD distribution by using the sample mean and log variance is made to inform the NF head of the MCNF method about the approximate location and spread of the prior distribution to assist in the reconstruction of the actual distribution of the prediction error ($\delta$), which is learnt from the data.
>
>
> > The training loss includes a temperature-scaled softmax to weigh observations, mitigating the effect of outliers. In the experiments, τ is set to a very large value (1e10), effectively giving equal weight to all observations. Why was this choice made?
>
> A: The proposed loss function was designed so that it falls back to the original Negative Log-Likelihood (NLL) for sufficiently large values of the $\tau$ parameter. For datasets containing no outliers, there is no reason to penalize observations differently, and this principle guided our approach to deal with all of the datasets taken from the UCI Benchmark. Romano’s synthetic dataset (Appendix B) was used as a counterfactual example to demonstrate the benefits of this modeling framework in mitigating the negative impact of the presence of outliers on the maximum likelihood approach. In these cases, $\tau$ also moderates the importance of highly informative priors, as is the case for the proposed MCNF. In Section C.2.2. of the Appendix (supplementary material), we argue that $\tau$ should be set by using a hold-out calibration set and discuss the effect of this hyperparameter on the performance metrics (coverage, quantile MAE, and prediction interval sizes) by analyzing a set of different $\tau$ values.
>
>
> > How does MCNF compare to other methods that also use normalizing flows for uncertainty quantification, but perhaps in a different manner ?
>
> A: The existing literature on the use of Normalizing Flows for uncertainty quantification can be grouped into two main categories:
>
> * full Bayesian and variational approaches [1-2] that extend their formalism using normalizing flows to model priors or, alternatively, as the variational family of distributions to enrich the modelling of the posterior distribution over the parameters, respectively. The full Bayesian approach often resorts to some modelling assumptions to preserve tractability, which may limit the expressiveness of the modelling framework. This is the case of [1] where authors hypothesize a Gaussian distribution of the prediction variable uncertainty. The variational approach deduced in [2] has the inherent limitations of Variational Autoencoders (VAEs) in approximating the true density.
> * the second category include approaches using Normalizing Flows as the probabilistic deep learning framework that enable describing the uncertainty in the form of a predictive distribution, $p(y|\mathbf{x})$ [3-5], characterizing the distribution of uncertain parameters on physical simulations [4], or using normalizing flows to accurately model the In-Distribution data, enabling Out-of-Distribution detection based on the estimated likelihood.
>
> These methods were left out of the related work as we mainly focused on methods that could be applied post-hoc, as this is one of the key requirements considered to propose MCNF. None of the reviewed methods enable this feature. Approach [5] shares with our method the use of Normalizing Flows to describe reconstruction (prediction) errors; however, the method is not post-hoc as its architecture is not independent from the data structure. It is also important to stress that propagating epistemic uncertainty through the prediction error is crucial to improve the calibration of the uncertainty quantification. To support this latter argument, we refer the reviewer to the table below, in which the results of using NF to model prediction errors without propagating the epistemic uncertainty are shown. It is noteworthy that when no MCD is used to resample prediction errors, the NF becomes significantly less calibrated in most of the datasets included in the UCI Benchmark. The Table below includes results from the MCNF as proposed in Equation 3 and the results for Normalizing Flows not using prediction error resampling, demonstrating that MCNF yields significantly higher coverage than Normalizing Flows without prediction error resampling.
>
> |Dataset|**method**| **mae_median_mu** | **mae_median_sd**| **mae_quant_mu**| **mae_quant_sd**| **coverage_mu**| **coverage_sd**|
> |---|---|---|---|---|---|---|---|
> |Abalone| mcnf| 0.099| 0.007| 0.499| 0.017| 0.886| 0.017|
> |Abalone| nf| 0.098| 0.005| 0.494| 0.039| 0.874| 0.023|
> |Boston| mcnf| 0.078| 0.009| 0.428| 0.037| 0.904| 0.043|
> |Boston| nf| 0.075| 0.008| 0.300| 0.046| **0.784**| 0.081|
> |Concrete| mcnf| 0.085| 0.012| 0.472| 0.032| 0.920| 0.021|
> |Concrete| nf| 0.084| 0.008| 0.352| 0.020| **0.814**| 0.038|
> |Protein| mcnf| 0.202| 0.009| 1.172| 0.059| 0.927| 0.006|
> |Protein| nf| 0.186| 0.008| 1.048| 0.088| **0.887**| 0.018|
> |Romano-OG| mcnf| 0.406| 0.206| 2.977| 0.395| 0.926| 0.014|
> |Romano-OG| nf| 0.496| 0.154| 2.886| 0.254| 0.912| 0.019|
> |Wave| mcnf| 0.002| 0.001| 0.013| 0.001| 0.938| 0.030|
> |Wave| nf| 0.002| 0.001| 0.006| 0.001| **0.807**| 0.149|
> |Super| mcnf| 0.103| 0.005| 0.566| 0.023| 0.912| 0.009|
> |Super| nf| 0.096| 0.004| 0.462| 0.022| **0.866**| 0.011|
>
>
> [1] B. Zhang, W. Sui, Z. Huang, M. Li, and M. Qi, "Normalizing flow based uncertainty estimation for deep regression analysis," Neurocomputing, vol. 585, p. 127645, 2024.
>
> [2] R. Selvan, F. Faye, J. Middleton, and A. Pai, "Uncertainty quantification in medical image segmentation with normalizing flows," in Proc. Int. Workshop on Machine Learning in Medical Imaging (MLMI), MICCAI, 2020.
>
> [3] R. Orozco, M. Louboutin, A. Siahkoohi, G. Rizzuti, T. van Leeuwen, and F. J. Herrmann, "Amortized normalizing flows for transcranial ultrasound with uncertainty quantification," in Proc. Med. Imaging with Deep Learning (MIDL), vol. 227, Jul. 10–12, 2024, pp. 332–349.
>
> [4] A. Dasgupta and Z. W. Di, “Uncertainty quantification for ptychography using normalizing flows,” arXiv preprint arXiv:2111.00745, 2021.
>
> [5] S. Benaïchouche, G. Morel, F. Rousseau, and R. Fablet, “Divergence-free continuous normalizing flows for uncertainty quantification,” preprint, Oct. 2022.
>
>
> > A strength of conformal prediction is that it enjoys coverage guarantees for the queried alpha. Does MCNF enjoy some theoretical properties (even under ideal assumptions)?
>
> A: Currently, we cannot make any claims about MCNF inheriting properties from the conformal prediction theoretical framework. However, the nature of the maximum likelihood principle supporting our MCNF method favours (although does not guarantee) the achievement of calibrated coverages, as maximizing the likelihood of the ground truth observations also maximizes the probability of building prediction intervals containing the latter. The experimental work carried out (Table 1, Figure 3) provides strong evidence that MCNF provides comparable miscalibration rates to those provided by the baseline methods based on conformal prediction (e.g. CQR, MCCP).

---

> > ### Author Response · Authors · 2025-08-05
> >
> > Dear reviewer 8wsk,
> >
> > We sincerely hope that our responses have addressed your concerns to your satisfaction. We truly value your feedback and would be glad to offer any additional clarification should you require them.
> >
> > Thank you for your time and consideration.
> >
> > Sincerely,
> > MCNF Authors

---

### Comment · Area_Chair_sekm · 2025-08-03
**Author-Reviewer Discussion**

Dear Reviewers,

The author-reviewer discussion period is now open and will continue until August 6. Please review the authors’ rebuttal to determine whether it adequately addresses your concerns.  If you have further questions or comments, engage with the authors by acknowledging that you’ve read their response and providing additional feedback as needed

Sincerely,

Your AC

---

### Note · Authors · 2025-08-16

We thank the reviewers and area chair for their thoughtful engagement. Following discussions, reviewers' concerns were resolved, and the novelty, technical soundness, and impact of Monte Carlo Normalizing Flow (MCNF) were acknowledged.

MCNF is a post-hoc uncertainty quantification (UQ) method for regression that combines Monte Carlo Dropout (MCD) to propagate epistemic uncertainty with normalizing flows (NF) to learn prediction error distributions, capturing both epistemic and aleatoric uncertainty. MCNF can be applied to pretrained models without retraining, yielding flexible predictive distributions that capture asymmetry and multimodality. Experiments on UCI benchmarks, synthetic multimodal data, and a real-world molecular dataset show that MCNF achieves superior coverage–interval tradeoffs compared to MCD, CQR, MCCP, and DQR.

**MCD statistics**. The mean and variance of MCD samples are used only to build the NF context; full distributional information is preserved via resampled prediction errors, enabling MCNF to recover complex distributions, including bimodality.

**Aleatoric uncertainty in MCD baselines**. Our MCD implementation follows common NeurIPS/ICML practice where the constant variance term is omitted. While this may understate aleatoric effects, our comparison remains fair against SoTA UQ methods. We commit to stating this adaptation explicitly in the final version.

**Benchmarks and Metrics**. We focus on interpretable metrics (coverage, calibration, interval width). While NLL has known limitations across datasets, we agree to add NLL in the final version. Our evaluation uses standard UCI datasets, conforming to prior work, and extends to multimodal synthetic and molecular datasets, highlighting MCNF’s robustness.

**Interval adaptivity**. Interval width alone is not meaningful; rather, what matters is adaptivity to the underlying distribution. MCNF achieves strong calibration with narrower intervals than conformal methods, avoiding over-conservatism.

**Computational complexity**. Inference cost scales linearly (with n_MCD + n_NF). Robustness experiments show stable performance; MCNF is faster than MCCP and comparable to CQR/DQR.

MCNF is a lightweight, post-hoc, distribution-free UQ method that models complex predictive distributions without retraining. By clarifying reviewer concerns and committing to clearer MCD baseline reporting, we believe that MCNF is a valuable UQ contribution with clear relevance for the NeurIPS community.

---

### Decision · Program_Chairs · 2025-09-17

**Decision:**

Accept (poster)

**Comment:**

This paper introduces a novel post-hoc method for uncertainty quantification (UQ) in deep regression models. To improve uncertainty estimation, the authors propose leveraging both the features and outputs of the base model (MCD) as contexts to help refine and scale the predictive distribution modeled by a Normalizing Flow. Experiments on several benchmark datasets across multiple metrics demonstrate that the proposed method outperforms baseline approaches.

The paper received four borderline accepts. Reviewers generally find the idea of contextualized Normalizing Flows innovative and the experimental results convincing. However, concerns were raised about the high inference cost, the effectiveness of the contextual information, lack of clarity in some technical details, and potentially unfair comparison with one of the baselines. The authors’ rebuttal is extensive and detailed, providing additional empirical evidence that alleviates most major concerns.